# Uni-RL: Unifying Online and Offline RL via Implicit Value Regularization

**Haoran Xu**[*1]    **Liyuan Mao**[*2]    **Hui Jin**[*3]
**Weinan Zhang**[2]    **Xianyuan Zhan**[4]    **Amy Zhang**[1]
[1]University of Texas at Austin    [2]Shanghai Jiao Tong University
[3]Nanyang Technological University    [4]Tsinghua University

## Abstract

The practical implementations of reinforcement learning (RL) often face diverse settings, such as online, offline, and offline-to-online learning. Instead of developing separate algorithms for each setting, we propose Uni-RL, a unified model-free RL framework that addresses all these scenarios within a single formulation. Uni-RL builds on the Implicit Value Regularization (IVR) framework (Xu et al., 2023) and generalizes its dataset behavior constraint to the constraint w.r.t. a reference policy, yielding a unified value learning objective for general settings. The reference policy is chosen to be the target policy in the online setting and the behavior policy in the offline setting. Using an iteratively refined behavior policy solves the over-conservative issue of directly applying IVR in the online setting, it provides an implicit trust-region style update through the value function while being off-policy. Uni-RL also introduces a unified policy extraction objective that estimates in-sample policy gradient using only actions from the reference policy. This not only supports various policy classes, but also theoretically guarantees less value estimation error and larger performance improvement over the reference policy. We evaluate Uni-RL on a range of standard RL benchmarks across online, offline, and offline-to-online settings. In online RL, Uni-RL achieves higher sample efficiency than both off-policy methods without trust-region updates and on-policy methods with trust-region updates. In offline RL, Uni-RL retains the benefits of in-sample learning while outperforming IVR through better policy extraction. In offline-to-online RL, Uni-RL beats both constraint-based methods and unconstrained approaches by effectively balancing stability and adaptability.

Code: https://github.com/ryanxhr/Uni-RL

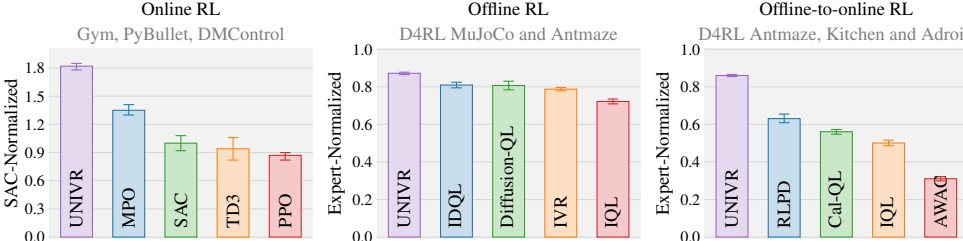

Figure 1: **Summary of results.** Aggregate mean performance across six common RL benchmarks and 23 environments with diverse characteristics (*e.g.,* observation and action spaces, task types, and offline data compositions). Error bars indicate the 95% stratified bootstrap confidence interval. UNIVR achieves competitive performance compared to state-of-the-art baselines across online, offline, and offline-to-online settings.

---

[*]Equal contribution. Correspondence to haoran.xu@utexas.edu.

39th Conference on Neural Information Processing Systems (NeurIPS 2025).

# 1 Introduction

Reinforcement learning (RL) has achieved impressive results across a wide range of domains, from playing complex games like Go (Mnih et al., 2013; Silver et al., 2017) to robotic manipulation tasks (Levine et al., 2016). However, the practical deployment of RL demands adaptability to diverse learning scenarios, such as online, offline, and offline-to-online settings. In the classical online RL paradigm, agents interact continuously with the environment, gaining experience and improving policy performance via trial-and-error approaches such as policy-gradient methods (Schulman et al., 2015, 2017) and actor-critic algorithms (Lillicrap et al., 2016; Fujimoto et al., 2018; Haarnoja et al., 2018). However, in safety-critical or high-cost domains, such as healthcare (Gottesman et al., 2018) and industrial control (Zhan et al., 2022, 2025), direct online exploration can be prohibitively risky or expensive, necessitating offline RL approaches that leverage pre-collected datasets without further environment interaction. Recent advances in offline RL have introduced conservative algorithms (Kumar et al., 2020; Fujimoto et al., 2019) and regularization-based methods (Wu et al., 2019; Xu et al., 2023) designed specifically to mitigate extrapolation errors arising from dataset distribution shifts. Moreover, many practical applications adopt an offline-to-online pipeline, initially leveraging offline datasets for safe and effective policy initialization, followed by online fine-tuning to explore more high-quality data to overcome the suboptimality of offline data (Nair et al., 2020; Lee et al., 2021b).

Existing RL algorithms are typically specialized for just one setting. For example, online algorithms (*e.g.*, PPO (Schulman et al., 2017), SAC (Haarnoja et al., 2018)) struggle with offline data due to distribution shift. Offline methods (*e.g.*, BCQ (Fujimoto et al., 2019), CQL (Kumar et al., 2020), IVR (Xu et al., 2023)) tend to be overly conservative or generalize poorly when given online access. Offline-to-online methods (*e.g.*, Off2On (Lee et al., 2021b), Cal-QL (Nakamoto et al., 2023)) often require special design choices, making them difficult to generalize to other settings. Designing different algorithms for each setting separately greatly limits the scalability and widespread adoption of RL in real-world applications. Therefore, a critical research question emerges:

*Can we design a scalable RL framework that unifies online, offline, and offline-to-online settings?*

**A Unified Reinforcement Learning Framework.** A desirable unified framework should fulfill several crucial properties: **(1) Adaptability**, seamlessly adapt to different settings, including online, offline, and offline-to-online, without changing the learning objective; **(2) Sample efficiency**, exhibit high sample efficiency, minimizing required interactions with the environment; and **(3) Scalability**, scale with growing volumes of data and learn effectively in large and complex environments. We consider the constraint optimization problem with a reference policy $\mu$:

$$\texttt{Uni-RL} \qquad \pi^* = \arg\max_\pi \mathbb{E}_{(s,a)\sim d^\pi} \left[ r(s,a) \right] \text{ s.t. } \mathbb{E}_{s\sim d^\pi} \left[ D_f \left( \pi(\cdot|s) \| \mu(\cdot|s) \right) \right] \leq \epsilon, \qquad (1)$$

where $D_f(p\|q) = \mathbb{E}_{x\sim q}[f(\frac{p(x)}{q(x)})]$ is the $f$-divergence (Boyd et al., 2004). By converting the constraint problem into an unconstrained, regularized one, we get a learning objective that imposes a policy-level value regularization by adding the $f$-divergence regularization term to the reward (Geist et al., 2019). This learning objective generalizes the Implicit Value Regularization (IVR) framework (Xu et al., 2023) from the dataset behavior constraint to the constraint w.r.t. the reference policy $\mu$, yielding **Unified RL (Uni-RL)**, a unified framework for general settings: For *offline RL*, $\mu$ is set to be the behavior policy $\pi_\mathcal{D}$ of the dataset. For *online RL*, we set $\mu$ to be the target policy $\bar{\pi}$ which is periodically or softly updated towards the current policy $\pi$. For *offline-to-online RL*, $\mu$ is set to be $\pi_\mathcal{D}$ in the offline pre-training stage while changing to $\bar{\pi}$ in the online stage. This framework is fully off-policy: offline RL can be considered as a one-step version of online RL (Brandfonbrener et al., 2021), and offline-to-online RL uses offline data to provide a good initialization for online RL.

One issue with Eq. (1) is the reliance on the probability distributions of $\pi$ and $\mu$, which limits the scalability of using advanced generative models such as diffusion and flow matching models (Song et al., 2020; Lipman et al., 2022) as the policy. However, we show later that by using techniques from duality (Xu et al., 2023; Sikchi et al., 2023), we can get the closed-form solution of the optimal policy $\pi^*$, and the value function of $\pi^*$ can be learned "in-sample" (using samples from the reference policy). By doing so, Uni-RL provides an implicit trust-region style update through the value function, and this implicit update results in an off-policy learning scheme that previous trust-region based methods could not achieve (Schulman et al., 2015, 2017). The implicit trust-region update not only increases sample-efficiency, but also solves the over-conservative issue when directly applying IVR in the

online setting. Furthermore, it enables smooth dataset constraint relaxation in offline-to-online RL, preventing early performance drops (Li et al., 2023). The in-sample learning scheme in Uni-RL also isolates the process of value learning and policy extraction, offering better learning stability. We further introduce a unified policy extraction objective that estimates in-sample policy gradient using only actions from the reference policy. This design scales successfully to more powerful policy classes beyond Gaussian distributions, and theoretically guarantees less value estimation error and larger performance improvement over the reference policy.

We evaluate Uni-RL on 6 widely used RL benchmarks and 23 environments across online, offline, and offline-to-online settings, achieving superior or competitive performance against state-of-the-art domain-specific baselines. In online RL, Uni-RL achieves higher sample efficiency than both off-policy methods without trust-region updates and on-policy methods with trust-region updates. In offline RL, Uni-RL retains the benefits of in-sample learning while outperforming IVR through better policy extraction. In offline-to-online RL, Uni-RL beats both constraint-based methods and unconstrained approaches by effectively balancing stability and adaptability.

## 2  Preliminaries

**Reinforcement Learning**   We consider the RL problem presented as a Markov Decision Process (MDP) (Sutton et al., 1998), which is specified by a tuple $\mathcal{M} = \langle \mathcal{S}, \mathcal{A}, \mathcal{P}, d_0, r, \gamma \rangle$. Here $\mathcal{S}$ and $\mathcal{A}$ are state and action space, $\mathcal{P}(s'|s, a)$ and $d_0$ denote transition dynamics and initial state distribution, $r(s, a)$ and $\gamma$ represent reward function and discount factor, respectively. The goal of RL is to find a policy $\pi(a|s)$ which maximizes expected return $J(\pi) = \mathbb{E}_\pi[\sum_{t=0}^\infty \gamma^t \cdot r(s_t, a_t)]$. Offline RL considers the setting where interaction with the environment is prohibited, and one needs to learn the optimal $\pi$ from a static replay buffer $\mathcal{D} = \{s_i, a_i, r_i, s_i'\}_{i=1}^N$. We also refer to $\mathcal{D}$ as the online replay buffer that is updated by filling in new transitions in the online or offline-to-online setting.

**Value functions and visitation distributions**   Let $V^\pi : \mathcal{S} \to \mathbb{R}$ and $Q^\pi : \mathcal{S} \times \mathcal{A} \to \mathbb{R}$ be the state and state-action value function of $\pi$, where $V^\pi(s) = \mathbb{E}_\pi[\sum_{t=0}^\infty \gamma^t r(s_t, a_t)|s_0 = s]$ and $Q^\pi(s, a) = \mathbb{E}_\pi[\sum_{t=0}^\infty \gamma^t r(s_t, a_t)|s_0 = s, a_0 = a]$. The visitation distribution $d^\pi$ is defined as $d^\pi(s, a) = (1 - \gamma) \sum_{t=0}^\infty \gamma^t \Pr(s_t = s, a_t = a \mid s_0 \sim d_0, \forall t, a_t \sim \pi(s_t), s_{t+1} \sim \mathcal{P}(s_t, a_t))$, which measures how likely $\pi$ is to encounter $(s, a)$ when interacting with the environment, averaging over time via $\gamma$-discounting. Let $V^*$, $Q^*$ and $d^*$ denote the value functions and visitation distribution corresponding to the regularized optimal policy $\pi^*$. We denote the empirical visitation distribution of $\mathcal{D}$ as $d^{\mathcal{D}}$ and the empirical behavior policy of $\mathcal{D}$ as $\pi_{\mathcal{D}}$, which represents the conditional distribution $p(a|s)$ observed in the dataset. Let $\mathcal{T}^\pi$ be the Bellman operator with policy $\pi$ such that $(\mathcal{T}^\pi Q)(s, a) := r(s, a) + \gamma \mathbb{E}_{s'|s,a} \mathbb{E}_{a' \sim \pi}[Q(s', a')]$ and $(\mathcal{T}^\pi V)(s) := \mathbb{E}_{a \sim \pi}[r(s, a) + \gamma \mathbb{E}_{s'|s,a}[V(s')]]$.

## 3  Uni-RL: Unified RL via Implicit Value Regularization

We give a detailed introduction of Uni-RL in this section. We begin by recalling how the dual form of Eq. (1) provides implicit value regularization, which is a unified, in-sample value learning objective. We then show a naïve extension of IVR to the online setting suffers from suboptimality, and address it with an iteratively improved behavior policy, resulting in an implicit trust-region style update. Finally, we propose a unified policy extraction method that is scalable and versatile to use across different policy classes. We theoretically prove that it enjoys lower value estimation error and larger performance improvement over the reference policy than previous policy extraction methods.

### 3.1  Towards Unified Value Learning

**Implicit Value Regularization.**   Note that the Lagrangian relaxation of Eq. (1) is equal to

$$\pi_{\text{IVR}}^* = \arg\max_\pi \ \mathbb{E}_\pi \left[ \sum_{t=0}^\infty \gamma^t \Big( r(s_t, a_t) - \alpha \cdot g\Big(\frac{\pi(a_t|s_t)}{\mu(a_t|s_t)}\Big) \Big) \right], \tag{2}$$

where $g(x) = f(x)/x$ is differentiable and satisfies $g(1) = 0$. Eq. (2) can be thought of as solving a *behavior-regularized* MDP problem with a modified reward function (Vieillard et al., 2020; Xu et al.,

2023). In this behavior-regularized MDP, the Bellman operator is changed to $\mathcal{T}_f^\pi$ such that

$$(\mathcal{T}_f^\pi)Q(s,a) := r(s,a) + \gamma\mathbb{E}_{s'|s,a}\left[V(s')\right]$$

$$V(s) = \mathbb{E}_{a\sim\pi}\left[Q(s,a) - \alpha \cdot g\left(\frac{\pi(a|s)}{\mu(a|s)}\right)\right].$$

Compared with the original Bellman operator $\mathcal{T}^\pi$, $\mathcal{T}_f^\pi$ is actually applying a value regularization to the $Q$-function. This transfers the greedy-max policy $\pi^*$ to a softened max (depending on $f$) over the reference policy $\mu$, which enables a scalable in-sample learning scheme[2].

**Lemma 1** (Results in IVR). *Using duality, the optimal value function $Q^*$ and $V^*$ can be solved by*

$$\min_V \ \mathbb{E}_{s\sim\mathcal{D},a\sim\mu}\left[V(s) + \alpha \cdot f_{\text{IVR}}\big((Q(s,a) - V(s))/\alpha\big)\right] \tag{3}$$

$$\min_Q \ \mathbb{E}_{(s,a,s')\sim\mathcal{D}}\left[\big(r(s,a) + \gamma V(s') - Q(s,a)\big)^2\right], \tag{4}$$

where $f_{\text{IVR}} = \exp(x)$ if $D_f$ is the KL divergence and $\mathbf{1}(x > -2)(x^2/4 + x + 1)$ if $D_f$ is the $\chi^2$ divergence. Note that Eq. (3) only uses samples from the reference policy distribution, without needing the knowledge of $\pi(a|s)$ and $\mu(a|s)$. This makes the usage of advanced generative models possible in the online setting, and avoids the need to use additional models to fit a dataset behavior distribution in the offline setting. We can recover IVR by setting $\mu$ to be $\pi_\mathcal{D}$, where offline actions serve as existing samples from $\pi_\mathcal{D}$.

**Does IVR work in the online setting?** We first examine whether IVR can be naturally extended to the online setting. In online IVR, the offline dataset will be the replay buffer collected so far. Unlike the offline setting, the replay buffer is collected by the learned policy that is periodically updated, which means it may contain a large portion of suboptimal or random data previously collected. Learning from a highly suboptimal dataset is known to be hard in offline RL due to the tendency to anchor the learned policy to the dataset behavior policy caused by regularization (Hong et al., 2023; Xu et al., 2025). In online IVR, the optimality of the regularized policy also highly depends on the replay buffer, and we need to break the regularization barrier to get the optimal value function and policy.

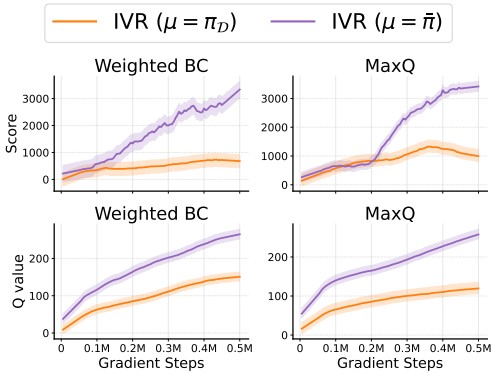

Figure 2: Using an **iteratively refined** behavior policy $\mu$ improves online IVR across different policy extraction methods.

One way to do so is by filtering the offline dataset iteratively according to the learned optimal policy at each iteration, and gradually optimizing towards the optimal policy using the filtered dataset, as done in Xu et al. (2025). However, this dataset filtering process is costly in the online setting. We thus propose a more lightweight option, where instead of iteratively refining the dataset, we maintain an iteratively refined reference policy $\bar{\pi}$, which is updated towards the current policy $\pi$ and improves beyond $\pi_\mathcal{D}$. This yields the learning objective in the online or offline-to-online setting.

$$\min_V \ \mathbb{E}_{s\sim\mathcal{D},a\sim\bar{\pi}}\left[V(s) + \alpha \cdot f_{\text{IVR}}\big([Q(s,a) - V(s)]/\alpha\big)\right] \text{ with } \bar{\pi} \leftarrow \lambda\pi + (1-\lambda)\bar{\pi}, \tag{5}$$

where $\lambda$ denotes the soft updating frequency, and if $\lambda = 1$, this amounts to assign $\bar{\pi}$ with the previous policy $\pi_{k-1}$, at iteration $k$. Using $\bar{\pi}$ as the reference policy gives a trust-region style update with an improved $\mu$, solving the over-constrained problem of directly applying IVR in the online setting. Figure 2 illustrates this, Uni-RL achieves higher Q values and better sample efficiency than IVR across different policy extraction methods. Since the trust-region update is implicitly imposed through the value learning, the algorithm remains off-policy. This improves sample-efficiency over previous trust-region methods, which are all on-policy (Schulman et al., 2015, 2017), and exhibits less value overestimation and more stability compared to existing off-policy methods (Fujimoto et al., 2018; Haarnoja et al., 2018; Abdolmaleki et al., 2018b). Furthermore, it enables smooth dataset constraint relaxation in offline-to-online RL, preventing the finetuned policy from early performance dropping (Nair et al., 2020; Nakamoto et al., 2023; Li et al., 2023).

---

[2]Here in-sample means samples from the reference policy $\mu$.

## 3.2 Towards Unified Policy Extraction

After introducing the unified value learning objective and demonstrating how Uni-RL addresses issues with IVR in the online setting, we now focus on providing a unified policy extraction scheme to effectively extract the best policy from the value functions learned with Uni-RL.

Note that in IVR (Eq. (2)), we have a closed-form solution for the ratio of the *optimal regularized policy* $\pi^*_{\text{IVR}}$ to the reference policy $\mu$ (Xu et al., 2023), which can be expressed as

$$w^*_{\text{IVR}}(s,a) = \frac{\pi^*_{\text{IVR}}(a|s)}{\mu(a|s)} = \max\left(0, (f')^{-1}\left((Q^*(s,a) - V^*(s))/\alpha\right)\right). \tag{6}$$

Previous works try to extract this policy by using either Forward KL or Reverse KL divergence.

**(1)** $D_{\text{KL}}\left(\pi^*_{\text{IVR}}\|\pi\right) \rightarrow$ **weighted BC**

$$\pi^* = \arg\max_\pi \mathbb{E}_{s\sim\mathcal{D}, a\sim\mu}\left[w^*_{\text{IVR}}(s,a) \cdot \log\pi(a|s)\right].$$

Using Forward KL divergence tends to be mode-covering, resulting in a weighted behavior cloning style loss where action is sampled from the reference policy. Although it queries the Q function with only in-sample actions (since Q is trained using actions from $\mu$), it prefers to cover all modes of the target distribution, including those with low probability mass (Park et al., 2024; Xu et al., 2025).

**(2)** $D_{\text{KL}}\left(\pi\|\pi^*_{\text{IVR}}\right) \rightarrow$ **MaxQ+BC**

$$\pi^* = \arg\max_\pi \mathbb{E}_{s\sim\mathcal{D}, a\sim\pi}\left[\log w^*_{\text{IVR}}(s,a) + \log\mu(a|s) - \log\pi(a|s)\right].$$

Using Reverse KL divergence gives a MaxQ+BC style loss (Tomar et al., 2020; Fujimoto and Gu, 2021; Mao et al., 2024b). Reverse KL divergence is mode-seeking, however, actions are sampled from the policy $\pi$, which are potentially out-of-distribution and causes over-estimation errors to the Q function. Futhermore, this loss needs an explicit $\mu(a|s)$, which might not be available in the offline setting, and $\mathbb{E}_{a\sim\pi}[\log\pi(a|s)]$ can be hard to estimate when $\pi$ is parameterized with generative models such as diffusion or flow matching models (Kong et al., 2023).

In summary, weighted BC has lower overestimation error and scales well across different settings and policy classes, while remaining mode-covering. In contrast, MaxQ+BC is mode-seeking but suffers from inaccurate gradient estimation and limited scalability. *Can we design a policy extraction method that shares the best of both worlds?*

**Policy extraction via In-sample Policy Gradient.** Our key intuition is that we still try to maintain the weighted BC style loss due to its stability and scalability, but we want to incorporate the first-order gradient information from policy gradient into the zero-order gradient induced by the weighted BC loss. To achieve that, we introduce In-sample Policy Gradient (**InPG**) where we ignore the original IVR weights $w^*_{\text{IVR}}(s,a)$ and instead think of those weights as learnable variables. In-sample PG projects the MaxQ gradient into the weighted BC gradient by learning the new **optimal weights** $w^*_{\text{InPG}}(s,a)$ such that

$$w^*_{\text{InPG}}(s,a) = \arg\min_{w\geq 0} \mathbb{E}_{s\sim\mathcal{D}}\left[\left(G_{\text{PG}} - G_{\text{BC}}(w)\right)^2\right]$$
$$\text{where} \quad G_{\text{PG}} = \nabla_\theta \mathbb{E}_{a\sim\pi_\theta}\left[Q(s,a)\right] \quad \text{and} \quad G_{\text{BC}}(w) = \mathbb{E}_{a\sim\mu}\left[w(s,a)\nabla_\theta\log\pi_\theta(a|s)\right]. \tag{7}$$

$G_{\text{BC}}(w)$ is the gradient vector of using weighted BC, we can modify the direction of this vector by changing the weight value on different $(s,a)$, and we want to find a non-negative $w$ metric to make the resulting vector stay as close as possible to the MaxQ gradient vector. Note that Eq.(7) is a constrained quadratic programming (QP) problem (Nocedal and Wright, 2006). Rather than solving it exactly using QP solvers (Stellato et al., 2020), we parameterize $w$ as a neural network and optimize it by minimizing the $L_2$ loss in Eq.(7) via gradient descent. We clip the $w$ at 0 during policy extraction to ensure weight non-negativity.

$$\pi^*_{\text{InPG}} = \arg\max_\pi \mathbb{E}_{s\sim\mathcal{D}, a\sim\mu}\left[w^*_{\text{InPG}}(s,a) \cdot \log\pi(a|s)\right]. \tag{8}$$

Using $w^*_{\text{InPG}}(s,a)$ as the BC weight enables better utilization of the Q function since using $w^*_{\text{IVR}}(s,a)$ cannot guarantee the policy is maximizing the Q function. It also makes the new weighted BC loss

more mode-seeking, enabling policy extraction under a support constraint since the BC actions are always within the support of the reference policy.

Theoretically, we find InPG could potentially enable a larger **performance improvement** than $\pi_{\text{IVR}}^*$ by increasing $J(\pi)$. In fact, InPG is optimizing towards the constraint learning objective from Eq.(1) while $\pi_{\text{IVR}}^*$ is the solution of Eq. (2), the relaxed Lagrangian of Eq. (1).

**Theorem 1.** *Define* $w_{\min} = \min_x w(x)$, $w_{\max} = \max_x w(x)$ *and* $Z = \mathbb{E}_{x \sim \mu}[w(x)] \in [w_{\min}, w_{\max}]$, *the* $f$-*divergence between* $\pi_{\text{InPG}}^*$ *and* $\mu$ *is bounded by*

$$f(\frac{w_{\min}}{Z}) \leq D_f(\pi_{\text{InPG}}^* \| \mu) \leq f(\frac{w_{\max}}{Z}).$$

**Theorem 2.** *There exists a range of* $w_{\min}$ *and* $w_{\max}$ *that satisfies* $J(\pi_{\text{InPG}}^*) \geq J(\pi_{\text{IVR}}^*)$.

We give the proof of these two theorems in Appendix A. These two theorems reveal that using in-sample policy gradient is actually finding the best policy (*i.e.,* maximizes the Q function) within a fixed constraint (depends on $w_{\min}$ and $w_{\max}$) rather than treating that constraint as a penalty, which is more aligned with the original optimization problem Eq. (1). This gives more freedom and allows for better policy extraction. The following theorem further demonstrates the necessity of applying the in-sample projection to the policy gradient, since directly using policy gradient has no improvement guarantee over the reference policy (also known to suffer from overestimation error in offline RL).

**Theorem 3.** *Define* $\pi_{\text{PG}}^* = \arg\max_\pi \mathbb{E}_{s \sim \mathcal{D}, a \sim \pi}[Q(s, a)]$, *there is no guarantee that* $J(\pi_{\text{PG}}^*) \geq J(\mu)$.

We use the `maze2d-umaze` dataset in D4RL (Fu et al., 2020) as an example to demonstrate the effect of using InPG. The Maze2D domain is a navigation task in which a 2D agent must reach a fixed goal location as quickly as possible; the action corresponds to the velocity along the $(x, y)$ axes. We visualize actions sampled from policies learned with IVR ($w_{\text{IVR}}^*(s, a)$) and Uni-RL ($w_{\text{InPG}}^*(s, a)$) in Figure 3. As shown, IVR tends to produce actions that are centered around the offline action distribution due to its mode-covering nature. In contrast, Uni-RL generates

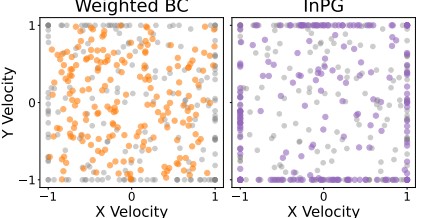

Figure 3: **InPG** discovers better actions from $\mathcal{D}$ (grey points) than **weighted BC**.

better actions that are near the boundaries of the action space by effectively using the value function. Intuitively, weighted behavior cloning can also exhibit mode-seeking behavior if appropriate weights are assigned to different actions. Notably, in-sample policy gradient does not require sensitive hyperparameter tuning like MaxQ+BC (Tarasov et al., 2024a; Park et al., 2025); it only requires setting a lower and upper weight threshold and we use the same value across all experiments.

**Scalable to advanced generative models.** In-sample policy gradient can easily scale to the usage of generative models like diffusion and flow matching models. Maximizing the log-likelihood objective in InPG is equivalent to minimizing the matching loss of different generative models (Song et al., 2021). For flow matching models, we have the following matching loss $L_{\text{Flow}}(\theta)$ (Lipman et al., 2022).

$$\mathbb{E}_{\substack{s \sim \mathcal{D}, a = x^1 \sim \mu, \\ x^0 \sim \mathcal{N}(0, I), \\ t \sim \text{Unif}([0,1])}} \left[ \| v_\theta(t, s, x^t) - (x^1 - x^0) \|^2 \right],$$

where $v_\theta(t, s, x)$ is a state-and-time dependent vector field with parameter $\theta$. For the use of diffusion models, we have the following matching loss $L_{\text{Diffusion}}(\theta)$ based on Ho et al. (2020).

**Algorithm 1** Unified Implicit Value Regularization

**Require:** $\mathcal{D}, \alpha$.
 1: Initialize $Q, V, \pi, \bar{\pi}$
 2: ▷ Unified value and policy learning
 3: **procedure** UNI-RL $(\pi, \mu, \mathcal{D})$
 4:     Sample transitions $(s, a, r, s') \sim \mathcal{D}$
 5:     Update $V$ by Eq.(3) with $a \sim \mu$, $Q$ and $\mathcal{D}$
 6:     Update $Q$ by Eq.(4) with $V$ and $\mathcal{D}$
 7:     Update $\pi$ by Eq.(7,8) with $a \sim \mu$, $Q$ and $\mathcal{D}$
 8: ▷ Offline training (offline and offline-to-online)
 9: **for** $t = 1, 2, \cdots, M$ **do**
10:     UNI-RL $(\pi, \pi_\mathcal{D}, \mathcal{D})$
11: ▷ Online finetuning (online and offline-to-online)
12: **for** $t = 1, 2, \cdots, N$ **do**
13:     Explore using $\pi$ and append $(s, a, r, s')$ to $\mathcal{D}$
14:     UNI-RL $(\pi, \bar{\pi}, \mathcal{D})$
15:     Update $\bar{\pi}$ by Eq.(5)

$$\mathbb{E}_{s \sim \mathcal{D}, a \sim \mu, \epsilon \sim \mathcal{N}(0, I), t \sim \text{Unif}([1, T])} \left[ \| \epsilon - \epsilon_\theta \left( \sqrt{\bar{\alpha}_t} a + \sqrt{1 - \bar{\alpha}_t} \epsilon, s, t \right) \|^2 \right]. \tag{9}$$

In summary, InPG provides a unified policy extraction method based on the weighted BC objective, which uses zero-order gradient while being mode-seeking to maximize Q values. InPG avoids value overestimation errors and scales to implicit behavior cloning through generative modeling, bringing better generalization. The full Uni-RL algorithm combines the unified policy extraction module with the unified value learning module. The pseudocode of Uni-RL is provided in Algorithm 1.

## 4 Prior Work

**Online RL.** Model-free online RL algorithms can be categorized into on-policy and off-policy methods. While on-policy methods only use experience from the current policy to perform updates (Kakade, 2001; Mnih et al., 2016), off-policy methods, in general, can utilize experience from any arbitrary policy (Watkins and Dayan, 1992; Haarnoja et al., 2018; Fujimoto et al., 2018). To stabilize training, previous methods (Schulman et al., 2015, 2017) add an explicit trust-region policy constraint that limits the derivation from the previous policy. However, this results in on-policy learning, while Uni-RL uses an implicit value regularization that is off-policy. Some works also use a weighted BC style loss (Abdolmaleki et al., 2018b,a; Oh et al., 2018; Tomar et al., 2020) in the policy extraction step. However, they don't impose regularization in the value learning. Their value learning and policy learning are coupled, causing the overestimation issue in the offline setting. Note that Uni-RL decouples the value and policy learning owing to the implicit value regularization.

**Offline RL.** To tackle the distributional shift problem, most model-free offline RL methods augment existing off-policy methods with a dataset behavior regularization term. One class of methods imposes behavior regularization explicitly as a divergence penalty (Wu et al., 2019; Kumar et al., 2019; Fujimoto and Gu, 2021) or intervene in the value learning to encourage staying near the behavioral distribution and being pessimistic about unknown state-action pairs (Nachum et al., 2019; Kumar et al., 2020; Kostrikov et al., 2021a; Xu et al., 2022; Wu et al., 2021; An et al., 2021). The other class of methods implicitly impose the behavior regularization through weighted behavior cloning (Kostrikov et al., 2021b; Xu et al., 2023), which filters useful actions to perform behavior cloning based on how advantageous they are. Compared with the first class, Uni-RL is inherently more stable due to its imitation-style policy update. Compared with the second class, Uni-RL is better at utilizing the value function to avoid suboptimal policy extraction.

**Offline-to-online RL.** Offline-to-online RL aims to overcome the suboptimality of pure offline learning by collecting more high-quality data with online fine-tuning (Lee et al., 2022; Zhang et al., 2023; Li et al., 2023). To prevent performance drop caused by distribution shift from the offline to online stage, previous methods impose regularization to stay close to the offline data or the offline pretrained policy during online fine-tuning Nair et al. (2020); Kostrikov et al. (2021b); Lee et al. (2022); Nakamoto et al. (2023); Zhao et al. (2022). However, these methods are over-conservative since the offline data could be highly suboptimal. Uni-RL solves this issue by enforcing an iteratively refined behavior policy initialized from an offline-pretrained policy, allowing us to gradually release the dataset constraint to continue improving throughout training. There are also some works that try to avoid the performance drop issue by ignoring the offline stage and learning from scratch using offline data (Song et al., 2022; Ball et al., 2023). However, these methods are less sample efficient since they lack the usage of pretrained value functions to perform effective exploration.

**Other prior works related to unified RL.** There are several other works that try to provide unification for general RL settings. Uni-O4 (Lei et al., 2023) provides unification of offline RL and offline-to-online RL based on on-policy policy gradient, but suffers from sample inefficiency compared to Uni-RL, which is off-policy. Policy Agnostic RL (Mark et al., 2024) provides unification for policy extraction methods, whereas Uni-RL also achieves a principled unification of value learning for general RL settings. Note that although some DICE-based methods (Lee et al., 2021a; Sikchi et al., 2023; Mao et al., 2024a) have a similar optimization objective with Uni-RL, they can't be unified in both the online and offline settings. Specifically, if we apply DICE-based methods to the online setting, sampling from the stationary distribution $d^\mu(s, a)$ of the reference policy is needed. This is intractable in the online setting, making DICE-based methods only work for the offline setting.

## 5 Experiments

**Online RL.** We first test Uni-RL in the online setting. We choose 8 environments from OpenAI Gym (Brockman et al., 2016), DeepMind Control Suite (Tassa et al., 2018), and PyBullet (Coumans and Bai, 2016), representing a large and diverse set of domains based on Box2D (Catto, 2011), MuJoCo (Todorov et al., 2012) and Bullet (Coumans et al., 2010) physics engines. To demonstrate the effectiveness of our method, we compare Uni-RL with several state-of-the-art model-free online RL algorithms including TD3 (Fujimoto et al., 2019), SAC (Haarnoja et al., 2018), PPO (Schulman

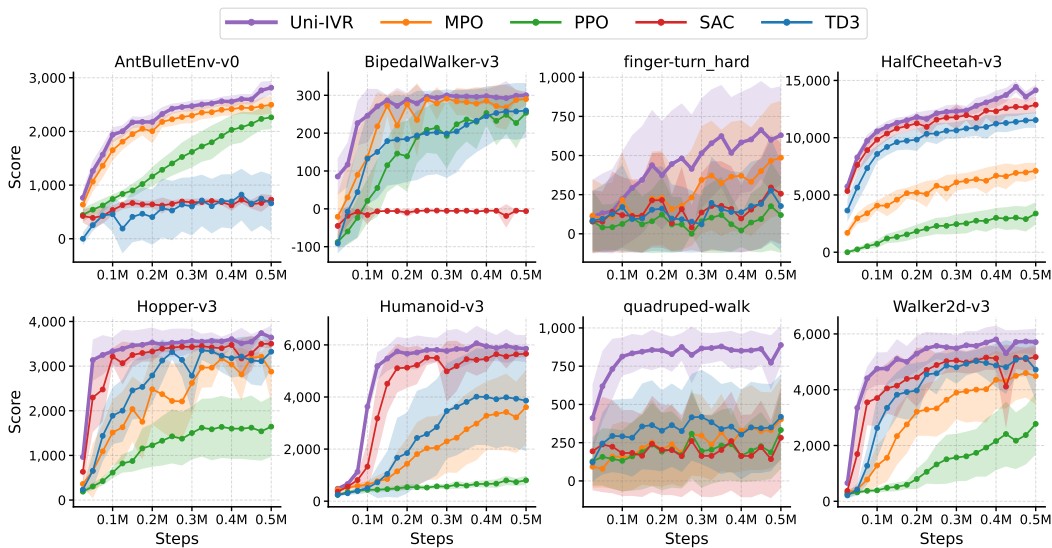

Figure 4: **Results in online RL.** Learning curves are plotted across 5 seeds with a smoothing window of 5000. Shading represents one standard deviation.

Table 1: **Results in offline RL**. Scores are averaged over the final 10 evaluations across 5 seeds with standard deviation reported, we highlight the best score in integer-level.

| Dataset | 10%BC | TD3+BC | CQL | IQL | IVR | Diffusion-QL | IDQL | Uni-RL (Gaussian) | Uni-RL (Diffusion) |
|---|---|---|---|---|---|---|---|---|---|
| halfcheetah-m | 42.5 | 48.3 | 44.0 ±0.8 | 47.4 ±0.2 | 48.3±0.2 | 51.1 ±0.5 | 51.0 | 49.4±0.2 | 58.0 ±0.6 |
| hopper-m | 56.9 | 59.3 | 58.5 ±2.1 | 66.3 ±5.7 | 75.5±3.4 | 90.5 ±4.6 | 65.4 | 99.5 ±1.1 | 101.3 ±0.6 |
| walker2d-m | 75.0 | 83.7 | 72.5 ±0.8 | 72.5 ±8.7 | 84.2±4.6 | 87.0 ±0.9 | 82.5 | 89.3±0.3 | 92.3 ±0.1 |
| halfcheetah-m-r | 40.6 | 44.6 | 45.5 ±0.5 | 44.2 ±1.2 | 44.8±0.7 | 47.8 ±0.3 | 45.9 | 45.3±0.3 | 48.4 ±0.9 |
| hopper-m-r | 75.9 | 60.9 | 95.0 ±6.4 | 95.2 ±8.6 | 99.7±3.3 | 101.3 ±0.6 | 92.1 | 101.1 ±2.7 | 101.3 ±2.1 |
| walker2d-m-r | 62.5 | 81.8 | 77.2 ±5.5 | 76.1 ±7.3 | 81.2±3.8 | 95.5 ±1.5 | 85.1 | 86.6±1.1 | 90.8 ±1.6 |
| halfcheetah-m-e | 92.9 | 90.7 | 90.7 ±4.3 | 86.7 ±5.3 | 94.0±0.4 | 96.8 ±0.3 | 95.9 | 94.2±0.6 | 97.3 ±0.6 |
| hopper-m-e | 110.9 | 98.0 | 105.4 ±6.8 | 101.5 ±7.3 | 111.8 ±2.2 | 111.1 ±1.3 | 108.6 | 111.0 ±0.6 | 111.2 ±0.3 |
| walker2d-m-e | 109.0 | 110.1 | 109.6 ±0.7 | 110.6 ±1.0 | 110.0±0.8 | 110.1 ±0.3 | 112.7 | 110.8±0.2 | 114.1 ±0.5 |
| antmaze-u | 62.8 | 78.6 | 84.8 ±2.3 | 85.5 ±1.9 | 92.2±1.4 | 93.4 ±3.4 | 94.0 | 94.1±1.6 | 98.1 ±1.8 |
| antmaze-u-d | 50.2 | 71.4 | 43.4 ±5.4 | 66.7 ±4.0 | 74.0±2.3 | 66.2 ±8.6 | 80.2 | 80.4±2.3 | 82.0 ±1.4 |
| antmaze-m-p | 5.4 | 10.6 | 65.2 ±4.8 | 72.2 ±5.3 | 80.2±3.7 | 76.6 ±10.8 | 84.5 | 86.0±2.6 | 88.5 ±3.1 |
| antmaze-m-d | 9.8 | 3.0 | 54.0±11.7 | 71.0 ±3.2 | 79.1±4.2 | 78.6 ±10.3 | 84.8 | 82.7±3.4 | 89.7 ±2.8 |
| antmaze-l-p | 0.0 | 0.2 | 38.4±12.3 | 39.6 ±4.5 | 53.2±4.8 | 46.4 ±8.3 | 63.5 | 59.9±2.9 | 68.6 ±3.6 |
| antmaze-l-d | 6.0 | 0.0 | 31.6±9.5 | 47.5 ±4.4 | 52.3±5.2 | 56.6 ±7.6 | 67.9 | 60.2±3.8 | 69.0 ±4.5 |

et al., 2017) and MPO (Abdolmaleki et al., 2018b). All baselines use a Gaussian policy, so we also use a Gaussian policy with Uni-RL for a fair comparison. The solid curve represents the average return, and the transparent shaded region represents the standard deviation. Each experiment was conducted over 5e5 training steps. According to the learning curves in Figure 2, Uni-RL achieves state-of-the-art performance and sample efficiency compared to the other four baseline algorithms, especially on challenging tasks in DMControl (*e.g.,* `finger-tune_hard` and `quadruped-walk`).

**Offline RL.** In the offline setting, we evaluate Uni-RL on the D4RL benchmark (Fu et al., 2020) and compare it with several state-of-the-art algorithms. For the evaluation tasks, we select MuJoCo locomotion tasks and AntMaze navigation tasks which require both locomotion and navigation. While MuJoCo tasks are popular in offline RL, AntMaze tasks are more challenging due to their stronger need for selecting optimal parts of different trajectories to perform stitching. For baseline algorithms, we selected state-of-the-art methods not only from traditional methods that use a Gaussian policy but also methods that use diffusion models. Gaussian-policy-based baselines include 10%BC (Chen et al., 2021), BCQ (Fujimoto et al., 2018), TD3+BC (Fujimoto and Gu, 2021), CQL (Kumar et al., 2020), IQL (Kostrikov et al., 2021a) and IVR (Xu et al., 2023). Diffusion-policy-based baselines include Diffusion-QL (Wang et al., 2023) and IDQL (Hansen-Estruch et al., 2023).

We implement Uni-RL with both Gaussian policy and Diffusion policy as in Eq.(9). The results in Table 1 show that Uni-RL with Gaussian policy already matches or outperforms most baseline algorithms, especially on MuJoCo medium, medium-replay datasets, and AntMaze datasets. The use

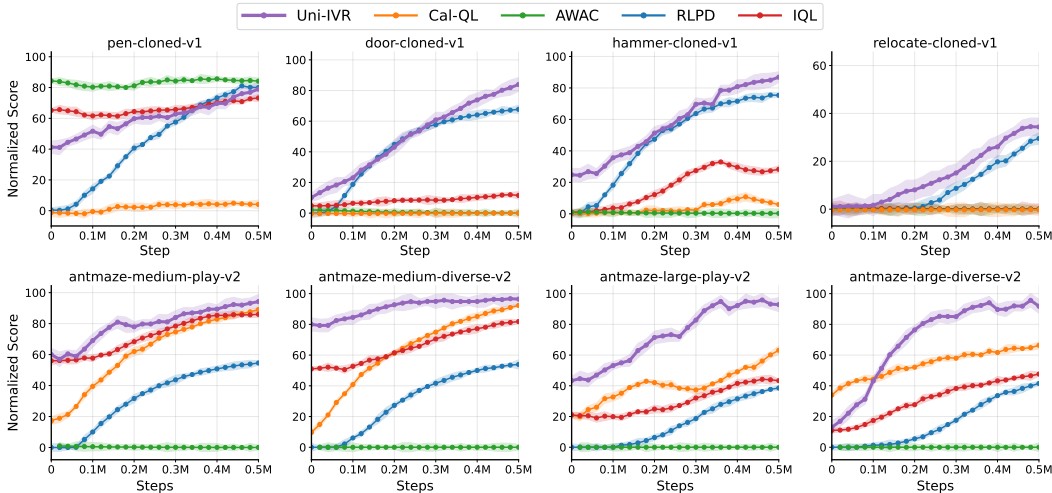

Figure 5: **Results in offline-to-online RL.** Learning Curves is plotted after 250k offline pretraining (5 seeds). Shading represents one standard deviation.

of diffusion models in Uni-RL further enhances its performance on tasks with multi-modal datasets, demonstrating the scalability of Uni-RL. Additionally, the consistently better performance of Uni-RL over IVR demonstrates the benefit of policy extraction using in-sample policy gradient.

**Offline-to-online RL.** In the offline-to-online setting, we conduct extensive experiments on AntMaze and Adroit tasks with D4RL datasets to demonstrate the stable, optimal policy learning and adaptability of Uni-RL. We compare Uni-RL with the following baselines: *(i)* AWAC (Nair et al., 2020): an offline-to-online method that learns the finetuning policy using AWR-style (Peng et al., 2019) policy loss. *(ii)* IQL (Kostrikov et al., 2021b): a SOTA offline RL approach based on weighted BC that can directly transfer to online finetuning. *(iii)* Cal-QL (Nakamoto et al., 2023): a SOTA offline-to-online approach specially designed based on CQL offline training. *(iv)* RLPD (Ball et al., 2023): a method that uses offline data to accelerate online training, it ignores the offline pretraining stage and learns from scratch. Note that we remove the high update-to-data trick in RLPD (*i.e.,* adding layer normalization and ensembles to the Q-function) for a fair comparasion.

Figure 5 shows that existing constraint-based approaches (IQL, AWAC) in most cases only marginally improve the offline pretrained policy, due to the over-conservatism introduced by the constraint w.r.t. the offline dataset. This is especially pronounced when the offline dataset or pretrained policy is highly-suboptimal such as in Adroit and Antmaze tasks. In contrast, Uni-RL enjoys stable initial finetuning and superior final performance owing to the iteratively refined policy regularization. Cal-QL is limited to CQL, making it hard to yield reasonable performance when the tasks are too difficult for CQL to obtain good results (*e.g.,* Adroit and Antmaze-large tasks). While RLPD achieves appealing results in some tasks, the sample efficiency is greatly limited in tasks with diverse or good offline data (*e.g.,* Antmaze tasks).

# 6   Limitations and Future Work

In this paper, we propose Uni-RL, a scalable framework that unifies different reinforcement learning settings, including online RL, offline RL, and offline-to-online RL. Uni-RL builds on the Implicit Value Regularization framework but generalizes the offline data constraint to a reference policy constraint, resulting in unified value learning and policy extraction objectives based on in-sample learning. Uni-RL is simple, effective, and scalable, and achieves superior performance across diverse RL settings. Uni-RL only considers the model-free setting and future directions include incorporating Uni-RL with model-based RL methods, and extending it to the LLM+RL setting. We believe that Uni-RL represents a concrete step toward building general and scalable off-policy RL algorithms.

## Acknowledgement

We thank members from MIDI lab for discussions on the method and feedback on the early draft of the paper. This work is partially supported by NSF 2340651, NSF 2402650, DARPA HR00112490431, and ARO W911NF-24-1-0193. This research used the computational cluster resource provided by the Texas Advanced Computing Center at UT Austin.

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

## A  Proof

**Theorem 1.** *Define $w_{\min} = \min_x w(x)$, $w_{\max} = \max_x w(x)$ and $Z = \mathbb{E}_{x \sim \mu}[w(x)] \in [w_{\min}, w_{\max}]$, the $f$-divergence between $\pi^*_{\text{InPG}}$ and $\mu$ is bounded by*

$$f(\frac{w_{\min}}{Z}) \leq D_f(\pi^*_{\text{InPG}} \| \mu) \leq f(\frac{w_{\max}}{Z}).$$

*Proof.* Remember that $\pi^*_{\text{InPG}}$ has the following expression:

$$\pi^*_{\text{InPG}}(a|s) = \frac{\mu(a|s)w^*(s,a)}{\sum_{a'} \mu(a'|s)w^*(s,a')}.$$

The $f$-divergence between $\pi$ and $\mu$ is defined as:

$$D_f(\pi^*_{\text{InPG}} \| \mu) = \sum_x \mu(x) f\left( \frac{\pi^*_{\text{InPG}}(x)}{\mu(x)} \right) = \sum_x \mu(x) f\left( \frac{\frac{w^*(x)\mu(x)}{Z}}{\mu(x)} \right)$$

$$= \sum_x \mu(x) f\left( \frac{w^*(x)}{Z} \right) = \mathbb{E}_{x \sim \mu}\left[ f\left( \frac{w^*(x)}{Z} \right) \right].$$

Given that $\omega(x)$ is bounded as:

$$w_{\min} \leq w(x) \leq w_{\max}, \quad \forall x,$$

we obtain the following bounds:

$$f\left( \frac{w_{\min}}{Z} \right) \leq \mathbb{E}_{x \sim \mu}\left[ f\left( \frac{w(x)}{Z} \right) \right] \leq f\left( \frac{w_{\max}}{Z} \right).$$

Thus, the $f$-divergence between $\pi^*_{\text{InPG}}$ and $\mu$ is bounded. $\qquad\square$

**Theorem 2.** *There exists $w_{\min}$ and $w_{\max}$ that satisfies $J(\pi^*_{\text{InPG}}) \geq J(\pi^*_{\text{IVR}})$.*

*Proof.* Remember that IVR considers a new MDP where the reward is augmented with a behavior regularization term $(r(s,a) = r(s,a) - \alpha g(\pi(a|s)/\mu(a|s)))$, so the $J(\pi)$ we consider here is equal to the value $V^\pi$. And an ideal algorithm wants to make sure that $V^\pi(s) \geq V^\mu(s)$, $s \sim \mathcal{D}$, i.e., safe policy improvement over the behavior policy over the replay buffer.

According to IVR,

$$V(s) = \mathbb{E}_{a \sim \pi}\left[ Q(s,a) - \alpha \cdot g\left( \frac{\pi(a|s)}{\mu(a|s)} \right) \right].$$

Since $\pi^*_{\text{InPG}}$ is the optimal solution of Eq. (7), i.e., the optimal behavior cloning weight that maximizes the Q function, we have

$$J(\pi^*_{\text{InPG}}) = \mathbb{E}_{s \sim \mathcal{D}, a \sim \pi^*_{\text{InPG}}}\left[ Q(s,a) - \alpha\, g\left( \frac{\pi^*_{\text{InPG}}(a|s)}{\mu(a|s)} \right) \right]$$

$$\geq \mathbb{E}_{s \sim \mathcal{D}, a \sim \pi^*_{\text{IVR}}}[Q(s,a)] - \alpha \mathbb{E}_{s \sim \mathcal{D}, a \sim \pi^*_{\text{InPG}}}\left[ g\left( \frac{\pi^*_{\text{InPG}}(a|s)}{\mu(a|s)} \right) \right]$$

$$= \mathbb{E}_{s \sim \mathcal{D}, a \sim \pi^*_{\text{IVR}}(a|s)}\left[ Q(s,a) - \alpha\, g\left( \frac{\pi^*_{\text{IVR}}(a|s)}{\mu(a|s)} \right) \right] + \alpha \mathbb{E}_{s \sim \mathcal{D}, a \sim \pi^*_{\text{IVR}}}\left[ g\left( \frac{\pi^*_{\text{IVR}}(a|s)}{\mu(a|s)} \right) \right]$$

$$- \alpha \mathbb{E}_{s \sim \mathcal{D}, a \sim \pi^*_{\text{InPG}}}\left[ g\left( \frac{\pi^*_{\text{InPG}}(a|s)}{\mu(a|s)} \right) \right]$$

$$= J(\pi^*_{\text{IVR}}) + \alpha \mathbb{E}_{s \sim \mathcal{D}, a \sim \pi^*_{\text{IVR}}}\left[ g\left( \frac{\pi^*_{\text{IVR}}(a|s)}{\mu(a|s)} \right) \right] - \alpha \mathbb{E}_{s \sim \mathcal{D}, a \sim \pi^*_{\text{InPG}}}\left[ g\left( \frac{\pi^*_{\text{InPG}}(a|s)}{\mu(a|s)} \right) \right].$$

To make $J(\pi_{\text{InPG}}^*) \geq J(\pi_{\text{IVR}}^*)$, one safe condition to satisfy is,

$$\mathbb{E}_{s\sim\mathcal{D}}\left[D_f(\pi_{\text{InPG}}^*(\cdot|s)\|\mu(\cdot|s))\right] = \mathbb{E}_{s\sim\mathcal{D},a\sim\pi_{\text{InPG}}^*}\left[g\left(\frac{\pi_{\text{InPG}}^*(a|s)}{\mu(a|s)}\right)\right]$$

$$\leq \mathbb{E}_{s\sim\mathcal{D},a\sim\pi_{\text{IVR}}^*}\left[g\left(\frac{\pi_{\text{IVR}}^*(a|s)}{\mu(a|s)}\right)\right] = \mathbb{E}_{s\sim\mathcal{D}}\left[D_f(\pi_{\text{IVR}}^*(\cdot|s)\|\mu(\cdot|s))\right].$$

Note that in Theorem 1 we have $f(w_{\min}/Z) \leq D_f(\pi_{\text{InPG}}^*\|\mu) \leq f(w_{\max}/Z)$. Let $f(w_{\max}/Z) \leq \mathbb{E}_s\left[D_f(\pi_{\text{IVR}}^*(\cdot|s)\|\mu(\cdot|s))\right]$ will satisfy this condition. In conclusion, the range of $w_{\min}$ and $w_{\max}$ to satisfy $J(\pi_{\text{InPG}}^*, \mu) \geq J(\pi_{\text{IVR}}^*, \mu)$ is

$$\frac{w_{\max}}{Z} \leq f^{-1}\left(\mathbb{E}_{s\sim\mathcal{D}}\left[D_f(\pi_{\text{IVR}}^*(\cdot|s)\|\mu(\cdot|s))\right]\right).$$

Note that some weaker conditions could also satisfy $J(\pi_{\text{InPG}}^*) \geq J(\pi_{\text{IVR}}^*)$. For example, we could choose $w_{\text{InPG}}^*$ to be close to $w_{\text{IVR}}^*$ such that $D_f(\pi_{\text{InPG}}^*(\cdot|s)\|\mu(\cdot|s)) \approx D_f(\pi_{\text{IVR}}^*(\cdot|s)\|\mu(\cdot|s))$. Also, in the worst case, setting $w_{\text{InPG}}^*(s,a) = w_{\text{IVR}}^*(s,a)$ we have $J(\pi_{\text{InPG}}^*) = J(\pi_{\text{IVR}}^*)$. $\qquad\square$

**Theorem 3.** *Define $\pi_{\text{PG}}^* = \arg\max_\pi \mathbb{E}_{s\sim\mathcal{D},a\sim\pi}[Q(s,a)]$, there is no guarantee that $J(\pi_{\text{PG}}^*) \geq J(\mu)$.*

*Proof.* Since $\pi_{\text{PG}}^*$ is the policy that maximizes $\mathbb{E}_{s\sim\mathcal{D},a\sim\pi}[Q(s,a)]$, we have

$$J(\pi_{\text{PG}}^*) = \mathbb{E}_{s\sim\mathcal{D},a\sim\pi_{\text{PG}}^*}\left[Q(s,a) - \alpha\, g\left(\frac{\pi_{\text{PG}}^*(a|s)}{\mu(a|s)}\right)\right]$$

$$\geq \mathbb{E}_{s\sim\mathcal{D},a\sim\mu}\left[Q(s,a)\right] - \alpha\mathbb{E}_{s\sim\mathcal{D},a\sim\pi_{\text{PG}}^*}\left[g\left(\frac{\pi_{\text{PG}}^*(a|s)}{\mu(a|s)}\right)\right]$$

$$= \mathbb{E}_{s\sim\mathcal{D},a\sim\mu}\left[Q(s,a) - \alpha\, g\left(\frac{\mu(a|s)}{\mu(a|s)}\right)\right] - \alpha\mathbb{E}_{s\sim\mathcal{D},a\sim\pi_{\text{PG}}^*}\left[g\left(\frac{\pi_{\text{PG}}^*(a|s)}{\mu(a|s)}\right)\right]$$

$$= J(\mu) - \alpha\mathbb{E}_{s\sim\mathcal{D},a\sim\pi_{\text{PG}}^*}\left[g\left(\frac{\pi_{\text{PG}}^*(a|s)}{\mu(a|s)}\right)\right] = J(\mu) - \alpha\mathbb{E}_{s\sim\mathcal{D}}\left[D_f\left(\pi_{\text{PG}}^*(\cdot|s)\|\mu(\cdot|s)\right)\right].$$

This inequality leverages the fact that $g(1) = 1 * f(1) = 0$. Since $D_f(\pi_{\text{PG}}^*(\cdot|s)\|\mu(\cdot|s)) \geq 0$, there is no guarantee that $J(\pi_{\text{PG}}^*) \geq J(\mu)$. $\qquad\square$

# B  Experimental Details

## B.1  Online RL experimental details

**Environments, Tasks, and Datasets**    We evaluate Uni-RL on 8 standard continuous control environments from OpenAI Gym, DeepMind Control Suite (DM Control), and PyBullet. These tasks vary in agent morphology, dimensionality, and dynamics complexity, ensuring a comprehensive evaluation of our method.

- `AntBulletEnv-v0`: It has a 28-dimensional state space consisting of joint positions, velocities, and body orientation features, and an 8-dimensional action space where each action dimension lies in [-1, 1]. The agent is a quadruped ant-like robot simulated using the PyBullet physics engine. The task requires learning stable and efficient forward locomotion under noisy and contact-rich dynamics.

- `BipedalWalker-v3`: This task has a 24-dimensional state space including hull angle, angular velocity, leg joint positions, and contact sensor readings, and a 4-dimensional continuous action space. The system represents a planar biped that must walk across varying terrain. The challenge lies in foot placement, balance control, and adapting to sparse footholds.

- `finger-turn-hard`: The environment has a 12-dimensional state space and a 6-dimensional action space. It features a robotic finger that must rotate an object to a specific target orientation. The dynamics are sensitive, the rewards are sparse, and the task demands fine motor control and long-horizon reasoning. This environment is from the DeepMind Control Suite.

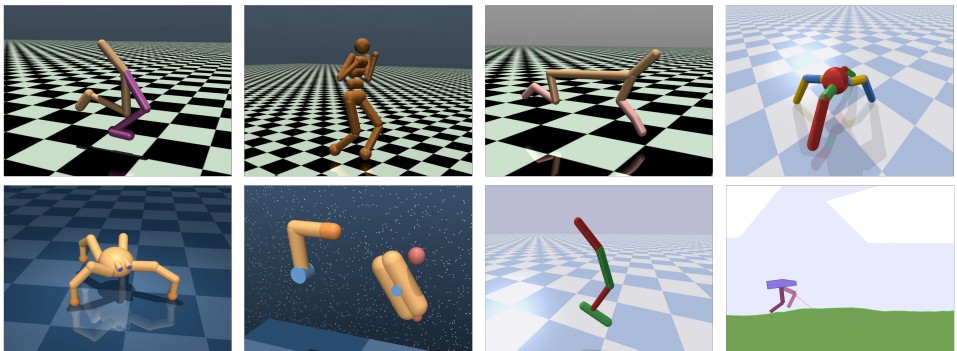

Figure 6: Tasks used in online RL.

- `HalfCheetah-v3`: The task includes a 17-dimensional state space representing joint positions and velocities, and a 6-dimensional action space. The system models a planar cheetah-like robot that learns to run forward as fast as possible. It is a widely used benchmark for evaluating the stability and efficiency of learned locomotion policies.

- `Hopper-v3`: The environment has an 11-dimensional state space and a 3-dimensional action space. It consists of a one-legged robot that must learn to hop forward without falling. The task is sensitive to small disturbances and evaluates learning in unstable, underactuated systems.

- `Humanoid-v3`: This is a high-dimensional task with a 376-dimensional state space and a 17-dimensional action space. The agent is a humanoid robot with 21 actuated joints and must learn to walk upright. The complexity of the dynamics and control makes it one of the most challenging continuous control benchmarks.

- `quadruped-walk`: The task includes a 78-dimensional state space representing joint and body kinematics and a 12-dimensional action space. The system models a quadrupedal robot using the DeepMind Control Suite. The goal is to learn stable walking behavior in a realistic 3D setting with proprioceptive sensing.

- `Walker2d-v3`: The environment has a 17-dimensional state space and a 6-dimensional action space. It simulates a planar bipedal robot that must learn to walk forward using two legs with multiple joints. This task provides a balanced challenge, commonly used to evaluate both stability and learning efficiency.

**Methods and Hyperparameters**     In all tasks, we computed the average mean returns over 10 evaluations every $5 \cdot 10^3$ training steps, across 5 different seeds.

For the network, we use 3-layer MLP with 256 hidden units and Adam optimizer (Kingma and Ba, 2015) with a learning rate of $1 \times 10^{-3}$ for both policy and value functions in all tasks. We also use a target network with soft update weight $5 \times 10^{-3}$ for Q-function. We clip the output of the weight function by $\max(w(x), 0)$ to ensure a non-negative BC weight. We use 0-1 normalization to the weight in each batch and then clip it to $[w_{min}, w_{max}]$ where we set $w_{min}$ to 0.1 and $w_{max}$ to 0.9 through all the datasets.

We implemented Uni-RL using PyTorch and ran it on all datasets. We followed the same reporting methods as mentioned earlier. In online experiments, we run baselines using the implementation from ACME (Hoffman et al., 2020)[3]. We use the reported hyperparameters in each paper.

In Uni-RL, we have two hyperparameters: regularization weight $\alpha$ and reference policy updating frequency $\lambda$. We search $\alpha$ over $[0.1, 0.5, 1.0, 2.0]$ and $\lambda$ over $[0.01, 0.05, 0.1, 0.2]$ The best value of $\alpha$ and $\lambda$ for all environments are listed in Table 3.

### B.2   Offline and Offline-To-Online RL experimental details

**Environments, Tasks, and Datasets**     In offline and offline-to-online, Uni-RL is evaluated on different kinds of datasets from various environments.

---

[3]https://github.com/google-deepmind/acme

For MuJoCo environments, we have the following datasets.

- `halfcheetah/hopper/walker2d-m` (medium): Collected by a policy with moderate performance, typically reaching around one-third of expert returns. These datasets represent structured but suboptimal behavior.

- `halfcheetah/hopper/walker2d-m-r` (medium-replay): Contains the replay buffer of the mediocre SAC policy. It includes a wide range of off-policy transitions, many of which are suboptimal or noisy.

- `halfcheetah/hopper/walker2d-m-e` (medium-expert): A 50-50 mixture of medium and expert trajectories. These datasets are designed to test whether algorithms can leverage near-optimal data when it is partially present.

The AntMaze environments involve a quadruped ant navigating through a 2D maze using sparse goal-based rewards. The agent has a 29-dimensional state space and an 8-dimensional action space, corresponding to joint positions, velocities, and target location encoding. The tasks are particularly challenging due to long-horizon planning and sparse supervision.

- `antmaze-u` (umaze): A small maze where the agent must reach a fixed goal location using sparse rewards. The environment is relatively easy due to short trajectories.

- `antmaze-u-d` (umaze-diverse): Similar to `umaze`, but with broader trajectory diversity collected from random exploration.

- `antmaze-m-p` (medium-play): A medium-sized maze where data is collected via a play policy. The task is harder due to longer horizons and sparse goal rewards.

- `antmaze-m-d` (medium-diverse): Features more diverse and noisy behavior than `medium-play`, increasing exploration coverage but decreasing consistency.

- `antmaze-l-p` (large-play): A large maze with random play data. The agent must navigate long distances, making the task especially difficult under sparse reward signals.

- `antmaze-l-d` (large-diverse): Similar to `large-play`, but with broader and more varied behavior. It is one of the most challenging offline datasets due to the size of the environment and variability in data.

The Adroit environments are high-dimensional dexterous manipulation tasks based on a 24-DoF Shadow Hand. Each environment has a 100-dimensional state space that includes joint angles, velocities, and object pose information, and a 24-dimensional continuous action space controlling finger joints. The cloned datasets are generated by behavior cloning from expert demonstrations and represent a moderate level of task success.

- `pen-cloned-v1`: The task requires rotating a pen to a target orientation using a 24-DoF anthropomorphic hand. The dataset is generated by cloning expert demonstrations. It is highly sensitive to precision and coordination.

- `door-cloned-v1`: A dexterous hand must unlock and open a door by grasping and rotating the handle. This task involves contact-rich control and precise force application, with cloned demonstrations as the data source.

- `hammer-cloned-v1`: The agent must use a hammer to drive a nail into a board. This involves both grasping and tool use, making it one of the most complex manipulation tasks in D4RL.

- `relocate-cloned-v1`: The goal is to pick up a ball and move it to a target location using precise grasping and positioning. The cloned dataset reflects human-like strategies but is difficult to exploit due to sparse rewards and high-dimensional dynamics.

**Methods and Hyperparameters** In Mujoco locomotion tasks, we computed the average mean returns over 10 evaluations every $5 \cdot 10^3$ training steps, across 5 different seeds. For Antmaze and Kitchen tasks, we calculated the average over 50 evaluations every $2 \cdot 10^4$ training steps, also across 5 seeds. We measure binary task success rates (in percentage) for AntMaze and normalized returns for Adroit, following the original evaluation scheme (Fu et al., 2020). Following previous research, we standardized the returns by dividing the difference in returns between the best and worst trajectories in MuJoCo tasks. In AntMaze tasks, we subtracted 1 from the rewards.

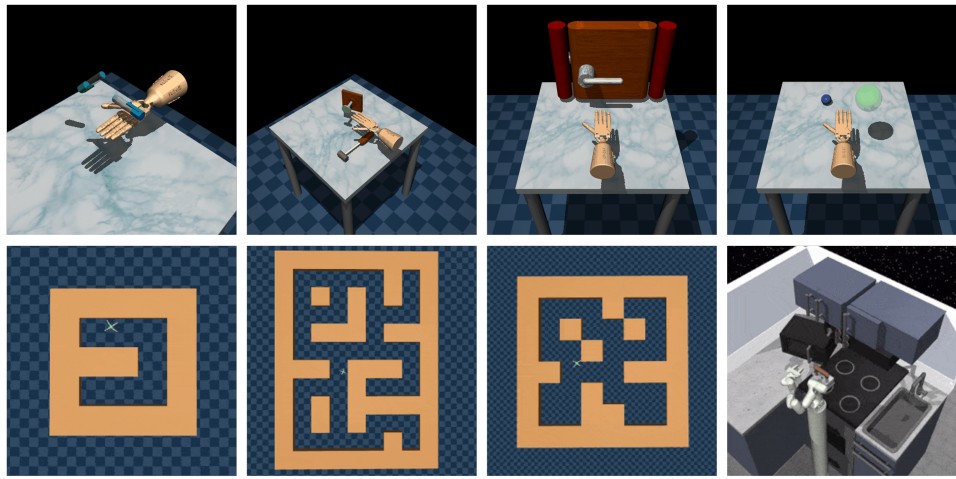

Figure 7: Tasks used in offline and offline-to-online RL.

Table 2: Hyperparameters for Uni-RL.

| Hyperparameter | Value |
| --- | --- |
| Learning rate | 0.0002 (offline), 0.001 (offline) |
| Optimizer | Adam |
| Gradient steps | 500000 |
| Minibatch size | 256 |
| MLP dimensions | $[256, 256, 256, 256]$ |
| Target network smoothing coefficient | 0.005 |
| Discount factor $\gamma$ | 0.99 |
| Diffusion steps (if used) | 10 |
| Regularization weight $\alpha$ | Tables 3 and 4 |
| $\bar{\pi}$ updating frequency $\lambda$ | Tables 3 and 4 |

For the network, we use 4-layer MLP with 256 hidden units and Adam optimizer (Kingma and Ba, 2015) with a learning rate of $2 \times 10^{-4}$ for both policy and value functions in all tasks. We also use a target network with soft update weight $5 \times 10^{-3}$ for Q-function. For Uni-RL with Diffusion models as the policy, the score network $\epsilon_\theta$ we used is based on a U-net architecture, which is fairly common in diffusion-based RL algorithms (Ajay et al., 2022; Mao et al., 2024b). We clip the output of the weight function by $\max(w(x), 0)$ to ensure a non-negative BC weight. We use 0-1 normalization to the weight in each batch and then clip it to $[w_{\min}, w_{\max}]$ where we set $w_{\min}$ to 0.1 and $w_{\max}$ to 0.9 through all the datasets.

We implemented Uni-RL using PyTorch and ran it on all datasets. We followed the same reporting methods as mentioned earlier. In offline experiments, baseline results for other methods were directly sourced from their respective papers. In offline-to-online experiments, we run baselines using the pytorch implementation from CORL (Tarasov et al., 2024b)[4]. Since CORL doesn't have the implementation of RLPD, we re-implement it in the codebase. We use the reported hyperparameters in each paper.

In Uni-RL, we have two hyperparameters: regularization weight $\alpha$ and reference policy updating frequency $\lambda$. We search $\alpha$ over $[0.1, 0.5, 1.0, 2.0]$ and $\lambda$ over $[0.01, 0.05, 0.1, 0.2]$ The best value of $\alpha$ and $\lambda$ for all datasets are listed in Table 4 and Table 5.

---

[4]https://github.com/corl-team/CORL

Table 3: Uni-RL hyperparameters in online RL.

| Env | $\alpha$ | $\lambda$ |
|---|---|---|
| AntBulletEnv-v0 | 1.0 | 0.1 |
| BipedalWalker-v3 | 1.0 | 0.1 |
| finger-turn-hard | 0.5 | 0.1 |
| quadruped-walk | 0.5 | 0.1 |
| HalfCheetah-v3 | 1.0 | 0.2 |
| Hopper-v3 | 1.0 | 0.2 |
| Humanoid-v3 | 1.0 | 0.2 |
| Walker2d-v3 | 1.0 | 0.2 |

Table 4: Uni-RL hyperparameters in offline RL.

| Env | $\alpha$ |
|---|---|
| halfcheetah-medium-v2 | 1.0 |
| hopper-medium-v2 | 1.0 |
| walker2d-medium-v2 | 1.0 |
| halfcheetah-medium-replay-v2 | 1.0 |
| hopper-medium-replay-v2 | 1.0 |
| walker2d-medium-replay-v2 | 1.0 |
| halfcheetah-medium-expert-v2 | 1.0 |
| hopper-medium-expert-v2 | 1.0 |
| walker2d-medium-expert-v2 | 1.0 |
| antmaze-umaze-v2 | 0.1 |
| antmaze-umaze-diverse-v2 | 2.0 |
| antmaze-medium-play-v2 | 0.1 |
| antmaze-medium-diverse-v2 | 0.1 |
| antmaze-large-play-v2 | 0.1 |
| antmaze-large-diverse-v2 | 0.1 |

Table 5: Uni-RL hyperparameters in offline-to-online RL.

| Env | $\alpha$ | $\lambda$ |
|---|---|---|
| pen-cloned-v1 | 1.0 | 0.01 |
| door-cloned-v1 | 2.0 | 0.01 |
| hammer-cloned-v1 | 2.0 | 0.01 |
| relocate-cloned-v1 | 2.0 | 0.01 |
| antmaze-medium-play-v2 | 0.5 | 0.01 |
| antmaze-medium-diverse-v2 | 0.5 | 0.01 |
| antmaze-large-play-v2 | 0.5 | 0.01 |
| antmaze-large-diverse-v2 | 0.5 | 0.01 |

