# OpenReview forum: "Uni-RL: Unifying Online and Offline RL via Implicit Value Regularization"
_NeurIPS.cc/2025/Conference — NeurIPS 2025 poster_

### Official Review · Reviewer_wWsA · 2025-06-30

**Clarity:** 3
**Significance:** 2
**Originality:** 2
**Rating:** 4
**Confidence:** 3

**Summary:**

This work proposes Uni-IVR, a single model-free algorithm that adapts to three settings --- offline, offline-to-online, and online learning. The key idea builds on the Implicit Value Regularization (IVR) framework and utilizes different reference policies depending on the settings: the behavior policy in the offline setting and a target policy in the online setting. Additionally, authors focus on providing a policy extraction scheme to effectively derive the best policy from the learned value function via Uni-IVR. The In-sample Policy Gradient (InPG) solves a constrained QP that determines weights using zero-order gradients, remains mode-seeking to maximize Q-values, and aligns a weighted BC loss with the true PG direction. In evaluation, Uni-IVR achieves higher sample efficiency and outperforms several SOTA algorithms in three settings.

**Questions:**

1. The related-work section cites recent algorithms such as Uni-O4 and DICE-based methods, yet Figure 5 benchmarks Uni-IVR only against Cal-QL, AWAC, RLPD, and IQL—methods published no later than 2023. To the best of my knowledge, BOORL [3], WSRL [4], and  O2SAC [5] are SOTA algorithms. Could you clarify whether Uni-IVR is expected to deliver superior sample efficiency, early-phase stability, and final performance when evaluated against these baselines?

2. Tables 4 and 5 list different hyperparameters for the Antmaze tasks: $\alpha$ = 0.1 or 2.0 in the pure-offline experiments, but $\alpha=0.5$ in the offline-to-online setting. 	Does this difference imply that a policy trained with the offline hyperparameter cannot be carried over directly into the online fine-tuning phase without re-tuning $\alpha$?

3. Lemma 1 appears to reproduce a result that was already proved in the original IVR paper (Xu et al., 2023). Please add an explicit citation and indicate the exact place in that paper where the derivation can be found.

[3] Hu, Hao, et al. "Bayesian design principles for offline-to-online reinforcement learning." arXiv preprint arXiv:2405.20984 (2024).

[4] Zhou, Zhiyuan, et al. "Efficient online reinforcement learning fine-tuning need not retain offline data." arXiv preprint arXiv:2412.07762 (2024).

[5] Luo, Qin-Wen, et al. "Optimistic critic reconstruction and constrained fine-tuning for general offline-to-online RL." Advances in Neural Information Processing Systems 37 (2024): 108167-108207.

**Ethical Concerns:**

["NO or VERY MINOR ethics concerns only"]

**Final Justification:**

The authors address the reviewer's concerns and provide more sufficient experiments.

**Limitations:**

In the Limitations section, the authors state only that Uni-IVR is restricted to the model-free setting. A more comprehensive ablation study is warranted. For instance, they could remove InPG (replacing it with vanilla BC or MaxQ + BC) and demonstrate robustness across a range of hyperparameter settings.

**Paper Formatting Concerns:**

None.

**Quality:**

3

**Strengths And Weaknesses:**

- Strengths
  1. While previous works in RL mainly focus on one of three settings, this work designs a scalable RL framework that unifies online, offline, and offline-to-online settings.
  2. InPG combines mode-seeking behavior with in-support actions, avoiding the over-estimation of Q-value while outperforming existing weighted-BC or MaxQ+BC. Also, Theorems 1-3 guarantee that the policy extraction with InPG improves performance and readily scales to generative models like diffusion and flow matching models.
  3. When comparing several algorithms in each RL setting, Uni-IVR achieves SOTA performance and sample efficiency in diverse datasets.


- Weaknesses
   1. Uni-IVR is closely related to IVR (Xu et al., 2023). The main difference in offline RL settings is the addition of the InPG method. In Table 1, I think IVR's performance is comparable to that of Uni-IVR (Gaussian). Additionally, baselines (BCQ, TD3+BC, CQL, and IQL) are not SOTA algorithms in offline RL. To make the empirical evaluation more convincing, Uni-IVR should be compared with more recent methods, such as ReBRAC [1] and QDQ [2].
   2. Lemma 1 states that the optimal value functions $Q^{\*}$ and $V^{\*}$  can be solved by Equations 3 and 4, respectively. In Equation 5, however, the authors replace the maximisation with an expectation over actions sampled from $\bar{\pi}$. Does this modified operator still converge to $V^{*}$?
   3. In Figure 5, learning curves are plotted after 250k offline pertaining. Yet this work claims that Uni-IVR (i) alleviates Q-value overestimation and (ii) avoids the performance drop typically seen at the start of offline-to-online finetuning. To substantiate these points, wouldn’t it be essential to plot the offline performance at the end of pre-training, alongside the subsequent finetuning curves?

[1] Tarasov, Denis, et al. "Revisiting the minimalist approach to offline reinforcement learning." Advances in Neural Information Processing Systems 36 (2023): 11592-11620.

[2] Zhang, Jing, et al. "Q-Distribution guided Q-learning for offline reinforcement learning: Uncertainty penalized Q-value via consistency model." Advances in Neural Information Processing Systems 37 (2024): 54421-54462.

---

> ### Author Rebuttal · Authors · 2025-07-31
>
> We thank the reviewer for their thoughful review of our work. We address the reviewers questions and concerns below:
>
>
> >The related-work section cites recent algorithms such as Uni-O4 and DICE-based methods, yet Figure 5 benchmarks Uni-IVR only against Cal-QL, AWAC, RLPD, and IQL—methods published no later than 2023. To the best of my knowledge, BOORL [3], WSRL [4], and O2SAC [5] are SOTA algorithms. Could you clarify whether Uni-IVR is expected to deliver superior sample efficiency, early-phase stability, and final performance when evaluated against these baselines?.
>
> To further compare the performance of Uni-IVR, we evaluate it against BOORL, WSRL, and O2SAC on AntMaze tasks, as BOORL and O2SAC did not conduct experiments on the Adroit datasets. Uni-IVR continues to demonstrate strong performance compared to these baselines.
>
> |   Offline-to-Online (200k online steps)    | Uni-IVR | BOORL | WSRL | O2SAC |
> | --------------- | ---- | --- | --- |--- |
> | antmaze-medium-play-v2  | 85.7  | **91.9**  | -| 78.8 |
> | antmaze-medium-diverse-v2  | **99.1**   | 84.1| -  |70.6|
> | antmaze-large-play-v2  | 85.5 | 55.9  | **96** | 44.2 |
> | antmaze-large-diverse-v2  | **98.1**  | 69.1  | 95 | 40.2 |
>
> >Tables 4 and 5 list different hyperparameters for the Antmaze tasks:
> alpha = 0.1 or 2.0 in the pure-offline experiments, but alpha=0.5 in the offline-to-online setting. Does this difference imply that a policy trained with the offline hyperparameter cannot be carried over directly into the online fine-tuning phase without re-tuning alpha?
>
> There is no guarantee that the best hyperparameters for the offline setting will be optimal for the off-to-online setting. Empirically, we find that using a less optimistic $\alpha$ leads to more stable value learning, which in turn benefits online fine-tuning.
>
>
> >Lemma 1 appears to reproduce a result that was already proved in the original IVR paper (Xu et al., 2023). Please add an explicit citation and indicate the exact place in that paper where the derivation can be found.
>
> Thank you for pointing that out. We would like to clarify that Lemma 1 is a result from the IVR paper and is not claimed as a novel contribution in our work (as indicated by labeling it as a lemma and explicitly stating it is from IVR). It is included solely for completeness and consistency in the presentation. The detailed derivation is provided in Appendix C in the IVR paper.
>
> >In the Limitations section, the authors state only that Uni-IVR is restricted to the model-free setting. A more comprehensive ablation study is warranted. For instance, they could remove InPG (replacing it with vanilla BC or MaxQ + BC) and demonstrate robustness across a range of hyperparameter settings.
>
> Due to space limit, please see the response to Reviewer HJMu for a comprehensive ablation study including different $\lambda$, using InPG vs weighted BC vs MaxQ, using Uni-IVR vs IVR.
>
>
> >Uni-IVR is closely related to IVR (Xu et al., 2023). The main difference in offline RL settings is the addition of the InPG method. In Table 1, I think IVR's performance is comparable to that of Uni-IVR (Gaussian). Additionally, baselines (BCQ, TD3+BC, CQL, and IQL) are not SOTA algorithms in offline RL. To make the empirical evaluation more convincing, Uni-IVR should be compared with more recent methods, such as ReBRAC [1] and QDQ [2].
>
> We compare Uni-IVR with ReBRAC and QDQ, and find that Uni-IVR achieves comparable performance to these advanced methods. However, we would like to emphasize that the key contribution of our work is **not** to outperform methods specifically designed for the offline setting. Rather, our goal is to propose a **general and unified framework** applicable across a wide range of RL settings, including but not limited to the offline scenario.
>
>
> |  Offline setting      | Uni-IVR | ReBRAC | QDQ |
> | --------------- | ---- | --- |--- |
> | halfcheetah-m  | 49.4  | 65.6  | 74.1  |
> | hopper-m  | 102.1  | 102  |  99.0  |
> | walker2d-m  | 89.3 | 82.5  |  86.9 |
> | halfcheetah-m-r  | 45.3 | 51  |  63.7  |
> | hopper-m-r  | 101.1  | 98.1  |  102.4  |
> | walker2d-m-r  | 86.6 | 77.3  |  93.2  |
> | halfcheetah-m-e  |  94.2 | 101.1 | 99.3  |
> | hopper-m-e  | 111.0 | 107  |  113.5  |
> | walker2d-m-e  | 110.8 | 111.6  |  115.9  |
> | antmaze-u  | 94.1 | 97.8  |  98.6  |
> | antmaze-u-d  | 80.4 | 88.3  | 67.8  |
> | antmaze-m-p  | 86.0 | 84.0  |  81.5  |
> | antmaze-m-d  | 82.7 | 76.3  |  85.4  |
> | antmaze-l-p  | 59.9 | 60.4  |35.6  |
> | antmaze-l-d  | 60.2 | 54.4 |  31.2  |
>
>
> >Lemma 1 states that the optimal value functions
> Q* and V* can be solved by Equations 3 and 4, respectively. In Equation 5, however, the authors replace the maximisation with an expectation over actions sampled from \hat{\pi}. Does this modified operator still converge to V*?
>
> Note that Lemma 1 shows that updating according to Equations (3) and (4) yields the optimal policy $\pi^{\ast}$ and value function $V^{\ast}$ that maximize the expected return **under the constraint induced by $\mu$**. In the online setting, we replace $\mu$ with a weighted mixture of $\pi^{\ast}$ and $\mu$, which ensures continual improvement of the reference policy. This allows $\pi^{\ast}$ to progressively improve toward the global optimum.
>
>
> >In Figure 5, learning curves are plotted after 250k offline pertaining. Yet this work claims that Uni-IVR (i) alleviates Q-value overestimation and (ii) avoids the performance drop typically seen at the start of offline-to-online finetuning. To substantiate these points, wouldn’t it be essential to plot the offline performance at the end of pre-training, alongside the subsequent finetuning curves?
>
> The offline performance at the end of pre-training is reflected in the initial point of the online learning curve (which is non-zero). Our method improves upon this initial performance without dropping down during online fine-tuning, supporting claim (2). Additionally, the high sample efficiency observed during online learning further supports claim (1).
>
>
> Please let us know if any further questions remain. We hope the reviewer can reassess our work with these clarifications.

---

> > ### Comment · Reviewer_wWsA · 2025-08-05
> >
> > Thank you for your response. However, the reviewer still has concerns.
> >
> > For example, the authors state that Uni-IVR outperforms additional baselines in the extra table for the offline-to-online setting. In the offline setting, the authors further state that *'' the key contribution of our work is not to outperform methods specifically designed for the offline setting. our goal is to propose a general and unified framework applicable across a wide range of RL settings, including but not limited to the offline scenario.''* However, in the submitted paper, the authors state Uni-IVR performs better than baselines. This appears to be a contradiction.
> >
> > The reviewer notes that existing algorithms such as IQL and ReBRAC are also applicable to offline, offline-to-online, and online RL, and when they were published, they demonstrated superior performance compared to prior methods. Moreover, these algorithms typically use the same hyperparameters in both offline and offline-to-online settings. While the reviewer agrees that there is no guarantee that the best hyperparameters for the offline setting will be optimal for the offline-to-online setting, the fact that Uni-IVR requires inconsistent hyperparameter choices across settings could be considered a limitation.
> >
> > In the offline-to-online setting, the WSRL algorithm is not compared on the medium dataset. Given that WSRL shows better performance than Uni-IVR on the large dataset, it is reasonable to expect it to also perform well on the medium dataset. In Figure 5, the reported performance appears high. However, unlike Figure 4, no statistical confidence intervals are provided, making it difficult to determine whether the performance difference is statistically significant.
> >
> > Furthermore, in the rebuttal for reviewer MCWA, the reviewer does not see evidence that Uni-IVR outperforms the original IVR in the offline setting. When comparing normalized scores including confidence intervals, all results except for hopper-m fall within the same range of the normalized scores.

---

> > > ### Author Response · Authors · 2025-08-07
> > >
> > > >This appears to be a contradiction
> > >
> > > We’d like to clarify that there is no contradiction in our claims. The goal of our work is clear: we propose a general, unified, and scalable framework—Uni-IVR—that works across different settings. To demonstrate the effectiveness of this framework, it is essential to compare it against several commonly used baselines under different settings, and we have done so. In the offline setting, Uni-IVR outperforms established methods such as TD3+BC, IQL, IVR, Diffusion-QL, and IDQL. Even when compared with more advanced algorithms mentioned by the reviewer, Uni-IVR achieves comparable performance. This does not contradict our claims; rather, it reinforces the effectiveness and necessity of Uni-IVR.
> > >
> > > In other words, since the goal is not to develop techniques with the aim to be specifically tailored for offline RL—such as the tricks used in BRAC or uncertainty quantification strategies in QDQ, we do not expect our framework to greatly outperform these offline RL algorithms. However, it turns out that our method still performs on par with these approaches, all while maintaining the flexibility to scale across different RL settings.
> > >
> > > >The reviewer notes that existing algorithms such as IQL and ReBRAC are also applicable to offline, offline-to-online, and online RL, and when they were published, they demonstrated superior performance compared to prior methods
> > >
> > > We respectfully disagree with the reviewer on this point. While IQL and ReBRAC can indeed be applied in the online setting, they do not represent the kind of scalable and effective unifed framework this paper aims to build.
> > >
> > > For IQL, its behavior in the online setting closely resembles that of online IVR, since both rely on in-sample learning. However, as demonstrated in our paper, naively extending IVR to the online setting leads to over-conservatism, which significantly degrades performance.
> > >
> > > As for ReBRAC, it lacks a trust-region-based value update in the online setting—unlike our proposed Uni-IVR. Without this crucial component, its behavior aligns more with SAC, which we have shown underperforms compared to Uni-IVR in the online regime.
> > >
> > > In summary, although these offline methods can technically be used in online learning, they fall short of the unified and scalable framework we target. This is precisely why we developed Uni-IVR—to address these limitations and provide a more effective solution across settings.
> > >
> > > >In the offline-to-online setting, the WSRL algorithm is not compared on the medium dataset. Given that WSRL shows better performance than Uni-IVR on the large dataset, it is reasonable to expect it to also perform well on the medium dataset.
> > >
> > > We respectfully disagree with the reviewer and believe this is a subjective interpretation. Uni-IVR outperforms WSRL on the antmaze-large-diverse task, while it underperforms on antmaze-large-play. Therefore, it is inaccurate to conclude that WSRL demonstrates better performance than Uni-IVR on the large dataset as a whole.
> > >
> > > Additionally, we would like to emphasize that these results are provided as additional experiments to further showcase Uni-IVR’s strong performance relative to a range of baselines. In the main paper, we have already presented comprehensive results where Uni-IVR clearly outperforms several strong baselines in the offline-to-online setting.
> > >
> > > >No statistical confidence intervals are provided, making it difficult to determine whether the performance difference is statistically significant.
> > >
> > > Thanks for pointing out that, we are sorry to miss that in the figure. Here are the reuslts in that figure, we will definitely add this to the final version.
> > >
> > > |  Offline-to-online      | Uni-IVR |
> > > | --------------- | ---- |
> > > | pen-cloned-v1  | 88.5±0.4  |
> > > | door-cloned-v1  | 93.3±1.9  |
> > > | hammer-cloned-v1  | 92.6±0.7   |
> > > | relocate-cloned-v1  | 38.5±0.3  |
> > > | antmaze-medium-play-v2  | 99.7±0.6   |
> > > | antmaze-medium-diverse-v2  | 99.1±0.6   |
> > > | antmaze-large-play-v2  | 92.5±1.3|
> > > | antmaze-large-diverse-v2  | 99.1±1.6   |
> > >
> > > >Furthermore, in the rebuttal for reviewer MCWA, the reviewer does not see evidence that Uni-IVR outperforms the original IVR in the offline setting.
> > >
> > > We respectfully disagree with this argument. Uni-IVR achieves improvements of over 5 points compared to IVR on 7 out of 15 tasks—including walker2d-medium, walker2d-medium-replay, hopper-medium, antmaze-umaze-diverse, antmaze-medium-play, antmaze-large-play, and antmaze-large-diverse. This represents a non-trivial gain on the D4RL benchmark.

---

> ### Comment · Reviewer_wWsA · 2025-08-08
>
> Thank you for your detailed response. I appreciate the clarification regarding the goal of Uni-IVR. However, I still have several concerns.
>
> While the authors claim that Uni-IVR outperforms strong baselines in both offline and offline-to-online settings, the baselines referenced (TD3+BC (2021), IQL (2021), Diffusion-QL (2023), and IDQL (2023)) have since been surpassed by several more recent SOTA algorithms. As such, I believe it is essential to compare Uni-IVR’s performance against a broader and more up-to-date set of baselines to validate its empirical contributions.
>
> Regarding the authors’ comment that *''it is inaccurate to conclude that WSRL demonstrates better performance than Uni-IVR on the large dataset as a whol''*: if the authors wished to make this argument, they should have included the WSRL scores for antmaze-medium-play and antmaze-medium-diverse in the first rebuttal. As a reviewer, I made a reasonable inference from the available results of the large dataset that WSRL likely outperforms Uni-IVR on the medium datasets. In the first rebuttal, the authors said that *''Uni-IVR continues to demonstrate strong performance compared to these baselines.''* As the authors themselves argue, it is inaccurate to conclude that Uni-IVR demonstrates better overall performance than other baselines without including the performance of WSRL on the medium dataset. Without those values, it is unclear why they were omitted, and the concern remains unaddressed.

---

> ### Author Response · Authors · 2025-08-09
>
> Thank you for your feedback.
>
> ### Regarding your first concern:
> We agree with the reviewer and do had included additional comparison results with more advanced methods in the rebuttal. As shown in the updated results, Uni-IVR maintains competitive performance. Note that Uni-IVR also consistently outperforms strong baselines in the online setting, which we believe represents a significant empirical contribution that should not be overlooked.
>
> ### Regarding your second concern:
> There may be some misunderstanding, our point is that one cannot conclude WSRL is superior to Uni-IVR when WSRL is better than Uni-IVR on one dataset while underperforming on the other one. We clarify that we only report results that are available in the original publications of the respective methods. For example, WSRL does not include results on the medium datasets, so their absence is not due to selective reporting. We are not cherry-picking results that favor Uni-IVR. Taken together, both the original empirical results and the newly added comparisons with several stronger baselines are fairly enough to demonstrate the effectiveness of Uni-IVR.

---

> > ### Comment · Reviewer_wWsA · 2025-08-09
> >
> > Thanks to the authors for the detailed rebuttals. All of the reviewer’s concerns have been addressed. I will raise my score to a positive score.
> >
> > I sincerely hope that the revision will include thorough comparisons with the most recent SOTA baselines.

---

### Official Review · Reviewer_MCWA · 2025-06-30

**Clarity:** 2
**Significance:** 2
**Originality:** 2
**Rating:** 4
**Confidence:** 4

**Summary:**

This paper proposes **Uni-IVR**, an extension of IVR that achieves strong performance across offline, off-to-on, and online reinforcement learning (RL) settings, without requiring substantial modifications between these modes. The main contributions can be summarized in two parts: **iterative refinement** and **in-sample policy gradient (InPG)**.

The **iterative refinement** component uses the target policy instead of the behavior policy to optimize the value function $V$ in online RL and in the online phase of off-to-on RL. The **InPG** component introduces a weight optimization scheme for weighted behavioral cloning (BC), aiming to align the Q-function maximization gradient with the weighted BC gradient.

Empirically, Uni-IVR demonstrates strong performance across all evaluated domains, with high sample efficiency and low sensitivity to hyperparameter tuning.

**Questions:**

1. Could you provide ablation results for some of the key hyperparameters or for different choices of $f$-divergence used in the algorithm?

2. What specific method of policy mixing is used in the iterative refinement process, parameter mixing or probability mixing? If it is parameter mixing, could you clarify in what way this contributes to the novelty of the method?

3. Given that IVR utilizes both KL divergence and $\chi^2$-divergence in separate algorithmic formulations, it would be helpful to clarify which divergence is used in Uni-IVR.

4. The use of 500K steps is an interesting design choice. However, it raises the question of whether training continues to improve beyond that point. Could you provide additional results or learning curves to illustrate this?

5. Would it be possible to include results using exactly the same hyperparameter settings as in the original IVR paper for the locomotion tasks in offline RL? This could help isolate the contribution of the proposed modifications.

**Ethical Concerns:**

["NO or VERY MINOR ethics concerns only"]

**Final Justification:**

The authors have adequately addressed the concerns I raised during the review period.

**Limitations:**

The authors are encouraged to explicitly discuss the limitations of their proposed algorithm, either in terms of its general applicability or in relation to the specific domains used in the experimental evaluation.

**Quality:**

3

**Strengths And Weaknesses:**

**Strengths**

1. InPG offers an intuitive and novel formulation of weighted behavioral cloning (BC).
2. The empirical results demonstrate strong performance, especially considering that Uni-IVR is applicable across diverse domains and operates as an in-distribution algorithm.
3. Uni-IVR requires minimal hyperparameter tuning and performs well with configurations close to standard defaults.
4. The theoretical support is thorough and well-presented.

**Weaknesses**

1. Some configurations and notations are ambiguous, including:
   - The method used to combine policies during iterative refinement. It is unclear whether this is done via parameter mixing or probability mixing.
   - The specific type of $f$-divergence employed. While IVR distinguishes between different divergences, the presented formulation appears similar to sparse Q-learning (SQL), which could benefit from clarification.
   - The type of noise scheduling used in diffusion-based cases. While VP scheduling is expected, cosine scheduling is also commonly applied and should be explicitly stated if used.

2. The iterative refinement component appears to be a straightforward application of the target policy method, which is already widely used in online RL algorithms such as TD3, as well as in some offline RL approaches. If the combination method corresponds to parameter mixing, the contribution may be limited to employing a larger mixing coefficient than the commonly used value of 0.005, which raises questions about the novelty of this aspect.

3. The paper does not include ablation studies, which makes it difficult to assess the individual contributions of each component.

4. If the second point holds, the overall algorithmic contribution may be somewhat limited.

---

> ### Author Rebuttal · Authors · 2025-07-31
>
> We thank the reviewer for their thoughful review of our work. We address the reviewers questions and concerns below:
>
>
> >Given that IVR utilizes both KL divergence and chi-divergence in separate algorithmic formulations, it would be helpful to clarify which divergence is used in Uni-IVR.
>
> Thanks for pointing out that, we use $\chi^2$-divergence throught the paper because SQL, which uses $\chi^2$-divergence, produces better and more stable results than EQL, which uses KL divergence. We will add this clarification in the revised paper.
>
>
> >Could you provide ablation results for some of the key hyperparameters in the algorithm?
>
> Due to space limit, please see the response to Reviewer HJMu for a comprehensive ablation study including different $\lambda$, using InPG vs weighted BC vs MaxQ, using Uni-IVR vs IVR.
>
>
> >What specific method of policy mixing is used in the iterative refinement process, parameter mixing or probability mixing? If it is parameter mixing, could you clarify in what way this contributes to the novelty of the method?
>
> We are using parameter mixing but do not claim the specific method of policy mixing as a contribution. Instead, the key contribution of this paper is **identifying the ineffectiveness of online IVR and proposing a lightweight solution that that not only addresses this issue but also generalize to other settings**. This solution further serves as a general framework applicable across various RL settings, which is a **non-trivial extension from the original offline IVR that no previous work has achieved**.
>
>
>
>
> >The use of 500K steps is an interesting design choice. However, it raises the question of whether training continues to improve beyond that point. Could you provide additional results or learning curves to illustrate this?
>
>
> Note that the key metric for evaluating online RL algorithms is the sample efficiency, i.e., achieve higher performance given limited interaction steps. We addiontially provide the results of training  1000K steps below to further show the performance. It can be seen that the improvement is limited with additional 500k steps, indicating that the learning curve is almost converged (we will add the learning curve to the revised paper).
>
>
> |   Online     | Uni-IVR (500k) | Uni-IVR (1000k) |
> | --------------- | ---- | --- |
> | Hopper-v3  | 3683±15  | 3691±4  |
> | HalfCheetah-v3  | 14305±25 | 15344±3  |
> | Humanoid-v3  | 5820±27 | 6034±6  |
> | Walker2d-v3  | 5940±16 | 6123±4  |
> | finger-turn_hard  | 625±80 | 654±22  |
> | quadruped-walk  | 924±14 | 945±12  |
> | AntBulletEnv-v0  |  2892±12 | 3021±9 |
> | BipedalWalker-v3  | 301±16 | 303±4  |
>
>
> >Would it be possible to include results using exactly the same hyperparameter settings as in the original IVR paper for the locomotion tasks in offline RL? This could help isolate the contribution of the proposed modifications.
>
> Note that we didn't change the hyperparameter settings of IVT when training Uni-IVR. The full comparision of Uni-IVR and IVR is shown below to justify the benefit of using InPG policy extraction.
>
> |  Offline setting      | Uni-IVR | IVR | Improvement |
> | --------------- | ---- | --- |--- |
> | halfcheetah-m  | 49.4±0.2  | 48.3±0.2  | +1.1  |
> | hopper-m  | 102.1 ±0.1 | 75.5±3.4  |  +26.6  |
> | walker2d-m  | 89.3±0.3 | 84.2±4.6  |  +5.1  |
> | halfcheetah-m-r  | 45.3±0.3 | 44.8±0.7  |  +0.7  |
> | hopper-m-r  | 101.1 ±1.7 | 99.7±0.3  |  +0.4  |
> | walker2d-m-r  | 86.6±1.1 | 81.2±3.8  |  +5.4  |
> | halfcheetah-m-e  |  94.2±0.6 | 94.0±0.4 | +0.2  |
> | hopper-m-e  | 111.0±0.6 | 111.0±0.2  |  =  |
> | walker2d-m-e  | 110.8±0.2 | 110.0±0.8  |  +0.8  |
> | antmaze-u  | 94.1±1.6 | 92.2±1.4  |  +1.9  |
> | antmaze-u-d  | 80.4±2.3 | 74.0±2.3  | +6.4  |
> | antmaze-m-p  | 86.0±2.6 | 80.2±3.7  |  +5.8  |
> | antmaze-m-d  | 82.7±3.4 | 79.1±4.2  |  +3.6  |
> | antmaze-l-p  | 59.9±2.9 | 53.2±4.8  |+6.7  |
> | antmaze-l-d  | 60.2±3.8 | 52.3±5.2  |  +7.9  |
>
>
> Please let us know if any further questions remain. We hope the reviewer can reassess our work with these clarifications.

---

> > ### Comment · Reviewer_MCWA · 2025-08-01
> >
> > Thank you for your detailed response. I appreciate the clarifications provided.
> >
> > However, two important points remain unresolved:
> >
> > 1. **Noise Scheduling in Uni-IVR**
> >    Please specify the type of noise scheduling (e.g., variance preserving (VP) or cosine) used in the diffusion model version of Uni-IVR. This detail is critical for reproducibility.
> >
> > 2. **Consistency in Hyperparameter Settings**
> >    While you state that Uni-IVR and IVR (SQL) share the same hyperparameters for offline RL, I observed notable discrepancies:
> >    - **Number of training steps:** 500K vs. 1M
> >    - **Learning rate:** 2e-4 vs. 3e-4 (locomotion) / 2e-4 (Antmaze)
> >    - **Model size:** [256, 256, 256, 256] vs. [256, 256]
> >
> >    To ensure a fair comparison, please **choose one of the two approaches**:
> >    - Match Uni-IVR’s settings to those of IVR (SQL), or
> >    - Match IVR (SQL)’s settings to those of Uni-IVR.
> >
> >    Then report the corresponding performance under the matched configuration.
> >    *(The difference in the $\alpha$ value for `antmaze-umaze-diverse` does not need to be changed, as its unusual behavior is understood.)*

---

> > > ### Author Response · Authors · 2025-08-03
> > >
> > > Thanks for pointing out that. Regarding your more questions, we have the replies below.
> > >
> > > 1. **Noise Scheduling in Uni-IVR**
> > >
> > > We consider the discrete and variance-preserving formulation, with linear noise scheduling as done in DDPM. We will make it more clear in the revised paper.
> > >
> > >
> > > 2. **Consistency in Hyperparameter Settings**
> > >
> > > Apologies for the earlier confusion—by same hyperparameters, we specifically refer to the crucial hyperparameter $\alpha$, which governs value function learning in IVR. Small differences in learning rate and model size had marginal impact on performance. To see it, we report results (averaged over 6 seeds) for both settings you mentioned: 1) Matching Uni-IVR’s hyperparameters to those of IVR, and 2) Matching IVR’s hyperparameters to those of Uni-IVR (using a re-run of the original IVR codebase).
> > >
> > > In the first case, the performance of Uni-IVR didn't drop, closely mirrors the original results. In the second case, we also didn't see performance improvement. Note that slight performance drop occurs in the AntMaze tasks, potentially due to platform differences—IVR is implemented in JAX, whereas Uni-IVR is implemented in PyTorch.
> > >
> > > In conclusion, slightly increasing model size and decreasing lr **doesn't** improve results in IVR, these variations had a far smaller effect on overall performance compared to the influence of the core hyperparameter $\alpha$, which remains the most critical factor for both IVR and Uni-IVR.
> > >
> > >
> > > |  matching Uni-IVR’s settings to those of IVR      | Uni-IVR | Uni-IVR (new config) |
> > > | --------------- | ---- | --- |
> > > | halfcheetah-m  | 49.4±0.2  | 49.8±0.5  |
> > > | hopper-m  | 102.1 ±0.1 | 101.5±0.2  |
> > > | walker2d-m  | 89.3±0.3 | 89.±0.1  |
> > > | halfcheetah-m-r  | 45.3±0.3 | 45.8±0.6  |
> > > | hopper-m-r  | 101.1 ±1.7 | 102.2±0.9  |
> > > | walker2d-m-r  | 86.6±1.1 | 87.4±0.8  |
> > > | halfcheetah-m-e  |  94.2±0.6 | 93.8±0.2 |
> > > | hopper-m-e  | 111.0±0.6 | 110.8±0.4  |
> > > | walker2d-m-e  | 110.8±0.2 | 110.2±0.3  |
> > > | antmaze-u  | 94.1±1.6 | 95.0±1.3  |
> > > | antmaze-u-d  | 80.4±2.3 | 82.1±2.4  |
> > > | antmaze-m-p  | 86.0±2.6 | 85.4±2.0  |
> > > | antmaze-m-d  | 82.7±3.4 | 82.0±2.7  |
> > > | antmaze-l-p  | 59.9±2.9 | 60.0±1.8  |
> > > | antmaze-l-d  | 60.2±3.8 | 59.7±2.3  |
> > >
> > >
> > > |  matching IVR’s settings to those of Uni-IVR      | IVR | IVR (new config) |
> > > | --------------- | ---- | --- |
> > > | halfcheetah-m  | 48.3±0.2  | 48.1±0.5  |
> > > | hopper-m  | 75.5±3.4 | 67.6±2.9  |
> > > | walker2d-m  | 84.2±4.6 | 83.8±3.8  |
> > > | halfcheetah-m-r  | 44.8±0.7 | 43.9±0.3  |
> > > | hopper-m-r  | 99.7±0.3 | 97.5±0.7  |
> > > | walker2d-m-r  | 81.2±3.8 | 80.0±2.3  |
> > > | halfcheetah-m-e  |  94.0±0.4 | 92.7±0.3 |
> > > | hopper-m-e  | 111.0±0.2 | 110.8±0.5  |
> > > | walker2d-m-e  | 110.0±0.8 | 109.7±0.5  |
> > > | antmaze-u  | 92.2± 1.6 | 92.8±1.1  |
> > > | antmaze-u-d  | 74.0± 2.3 | 67.1±2.1  |
> > > | antmaze-m-p  | 80.2± 3.7 | 72.4±3.1  |
> > > | antmaze-m-d  | 79.1± 4.2 | 73.0±3.5  |
> > > | antmaze-l-p  | 53.2± 4.8 | 50.0±4.0  |
> > > | antmaze-l-d  | 52.3± 5.2 | 51.2±4.3  |

---

> > > > ### Comment · Reviewer_MCWA · 2025-08-04
> > > >
> > > > I appreciate the authors’ efforts to address all of my questions and will adjust my rating accordingly.
> > > >
> > > > I sincerely hope that all the new results will be thoroughly incorporated into the revision.

---

> > > > > ### Author Response · Authors · 2025-08-05
> > > > >
> > > > > We sincerely appreciate you raising the score and offering insightful suggestions that significantly enhanced our paper.

---

### Official Review · Reviewer_HJMu · 2025-07-01

**Clarity:** 2
**Significance:** 3
**Originality:** 2
**Rating:** 4
**Confidence:** 4

**Summary:**

This paper proposes Uni-IVR, a single reinforcement learning framework designed to operate across online, offline, and offline-to-online settings. The method builds upon the Implicit Value Regularization (IVR) framework by generalizing its dataset-based constraint to a more flexible 'reference policy' constraint, which adapts to each learning paradigm. The paper also introduces a novel policy extraction method, In-sample Policy Gradient (InPG), to improve policy updates.

**Questions:**

1. Could you provide a full comparison of environmental returns for Uni-IVR vs. IVR across all online environments?
2. To validate the contribution of InPG, could you provide a comprehensive comparison of policy extraction methods (the proposed InPG vs. baselines like weighted BC and MaxQ) across all tasks in all three settings (online, offline, and offline-to-online)?
3. There is a significant discrepancy in the λ hyperparameter values between online (0.1–0.2) and offline-to-online (0.01) settings, as shown in Tables 3 and 5. This raises questions about the sensitivity of the method to this hyperparameter. Could you provide an ablation study that shows how performance varies with λ in each setting?
4. you explain how Uni-IVR leverages Q and V to guide training in these types of generative models and achieve the results shown in Table 1?
5. To support the claim of scalability, could you please add the results for a Flow-matching policy to Table 1, as was done for the Diffusion policy?
6. Could you provide the missing implementation details for InPG, including the architecture of the network used to learn the weights $w$, its optimization schedule, and an analysis of its training time compared to Weighted BC and MaxQ?
7. why Q-values in Figure 2 are used as indicator of performance?
8. In line 298-299, author stated that “DICE-based methods only work for the offline setting”. Can you justify the reason why they can not be applied by the reference policy?

**Ethical Concerns:**

["NO or VERY MINOR ethics concerns only"]

**Final Justification:**

The authors have addressed nearly all of my concerns; however, I still have some reservations regarding the implementation and reproducibility. The rating would increase if the source code were made available in the supplementary material.

**Limitations:**

The authors did not explicitly mention the societal impact of their work, but I do not perceive any negative societal implications.

**Quality:**

2

**Strengths And Weaknesses:**

**Strengths:**
- The paper tackles the important and ambitious goal of unifying RL across different learning paradigms (online, offline, offline-to-online) with a single, coherent algorithm.
- The approach is a well-motivated extension of the established IVR framework, providing a solid theoretical foundation for its value learning component.
- The experimental results are comprehensive and consistently demonstrate that Uni-IVR is competitive with or superior to a range of baseline methods across all three settings.

**Weaknesses:**
- The core value learning objective is a direct and incremental extension of the IVR framework. Moreover, the paper only compare the improvement in Value learning objective in one task with no name (Figure 2).
- The novel policy extraction method InPG lacks a comprehensive evaluation, only demonstrated primarily on a single environment (Figure 3), which raises concerns about the generalizability of the results. Moreover, the absence of supplementary source code makes it difficult to check the effective of this novel method.
- Lack of description about how to build and train $w$ in the implementation. Moreover, computational cost analysis for the InPG module are missing.
- The paper proposes two alternative objectives that enable the use of Flow-Matching and Diffusion models instead of traditional Gaussian policies. However, if I understand correctly, these objectives treat all actions from the dataset uniformly and do not incorporate information from the Q and V functions.
- in Figure 2, the authors provide Q-curves to illustrate higher performance; however, Q-learning methods (IQ-learn [1] for example) can produce high Q-values without corresponding performance gains. There are no explanation for these Q value results.


**Minors:**
- Figure 2 uses terms like MaxQ and Weighted BC without prior definition.
- Two provided learning rates in Table 2 are for offline.

[1] Garg, Divyansh, et al. "Iq-learn: Inverse soft-q learning for imitation." Advances in Neural Information Processing Systems 34 (2021): 4028-4039.

---

> ### Author Rebuttal · Authors · 2025-07-31
>
> We thank the reviewer for their time, effort, and constructive comments on our paper. We believe there may be some misunderstandings, which we address in detail below:
>
>
> >The core value learning objective is a direct and incremental extension of the IVR framework. Moreover, the paper only compare the improvement in Value learning objective in one task with no name (Figure 2).
>
> Note that although the value learning objective appears similar to the IVR objective, extending IVR to the online setting is non-trivial. The key contribution of this paper lies in **identifying the ineffectiveness of online IVR and proposing a lightweight solution that that not only addresses this issue but also generalize to other settings**. This fix naturally leads to a unified framework applicable across various RL settings.
>
> Figure 2 presents results on Hopper-v3. We apologize for omitting this information in the paper and will add it in the revised version. The observed behavior in Hopper-v3 is representative of all online environments; thus, we selected it as an illustrative example. For completeness, we provide the full performance comparison between Uni-IVR and IVR below. Note that IVR here uses the same policy extraction method (InPG) as Uni-IVR to ensure a fair comparison.
>
> | Online setting       | Uni-IVR | IVR |
> | --------------- | ---- | --- |
> | Hopper-v3  | 3683±15  | 2034±8  |
> | HalfCheetah-v3  | 14305±25 | 6743±28  |
> | Humanoid-v3  | 5820±27 | 3423±15  |
> | Walker2d-v3  | 5940±16 | 2334±19  |
> | finger-turn_hard  | 625±80 | 195±15  |
> | quadruped-walk  | 924±14 | 302±23  |
> | AntBulletEnv-v0  |  2892±12 | 2083±11 |
> | BipedalWalker-v3  | 301±16 | 71±24  |
>
> >The novel policy extraction method InPG lacks a comprehensive evaluation, only demonstrated primarily on a single environment (Figure 3), which raises concerns about the generalizability of the results. Moreover, the absence of supplementary source code makes it difficult to check the effective of this novel method.
>
> We respectfully disagree with the reviewer. As mentioned, Figure 3 is intended solely as an illustrative example to convey the motivation and intuition behind using InPG. The actual impact of InPG can be inferred from Table 1 in the paper by comparing Uni-IVR and IVR in the offline setting, where the value learning components are identical. For clarity, we include the relevant results here for easier reference.
>
> |  Offline      | Uni-IVR | IVR | Improvement |
> | --------------- | ---- | --- |--- |
> | halfcheetah-m  | 49.4±0.2  | 48.3±0.2  | +1.1  |
> | hopper-m  | 102.1 ±0.1 | 75.5±3.4  | +26.6  |
> | walker2d-m  | 89.3±0.3 | 84.2±4.6  | +5.1  |
> | halfcheetah-m-r  | 45.3±0.3 | 44.8±0.7  | +0.7  |
> | hopper-m-r  | 101.1 ±1.7 | 99.7±0.3  | +0.4  |
> | walker2d-m-r  | 86.6±1.1 | 81.2±3.8  | +5.4  |
> | halfcheetah-m-e  |  94.2±0.6 | 94.0±0.4 | +0.2  |
> | hopper-m-e  | 111.0±0.6 | 111.0±0.2  | =  |
> | walker2d-m-e  | 110.8±0.2 | 110.0±0.8  | +0.8  |
> | antmaze-u  | 94.1±1.6 | 92.2±1.4  | +1.9  |
> | antmaze-u-d  | 80.4±2.3 | 74.0±2.3  | +6.4  |
> | antmaze-m-p  | 86.0±2.6 | 80.2±3.7  | +5.8  |
> | antmaze-m-d  | 82.7±3.4 | 79.1±4.2  | +3.6  |
> | antmaze-l-p  | 59.9±2.9 | 53.2±4.8  | +6.7  |
> | antmaze-l-d  | 60.2±3.8 | 52.3±5.2  | +7.9  |
>
> In the online and offline-to-online setting, to validate the contribution of InPG, we provide the comparision of using InPG vs MaxQ and weighted BC as follows.
>
> |  Online      | Uni-IVR (InPG) | Uni-IVR (weighted BC) | Uni-IVR (MaxQ) |
> | --------------- | ---- | --- |--- |
> | Hopper-v3  | 3683±15  | 3234±34  | 3157±26  |
> | HalfCheetah-v3  | 14305±25 | 11334±12  | 12436±15  |
> | Humanoid-v3  | 5820±27 | 4845±15  | 4934±13  |
> | Walker2d-v3  | 5940±16 | 5134±8  |5310±34  |
> | finger-turn_hard  | 625±80 | 434±23  | 445±32  |
> | quadruped-walk  | 924±14 | 832±15  |781±24  |
> | AntBulletEnv-v0  |  2892±12 | 2245±17 | 2467±23  |
> | BipedalWalker-v3  | 301±16 | 296±24  |189±12  |
>
>
> |  Offline-to-online      | Uni-IVR (InPG) | Uni-IVR (weighted BC) | Uni-IVR (MaxQ) |
> | --------------- | ---- | --- |--- |
> | pen-cloned-v1  | 88.5±0.4  |80.6±0.5  | 65.5±0.6  |
> | door-cloned-v1  | 93.3±1.9  | 80.3±1.2  | 11.6±1.4  |
> | hammer-cloned-v1  | 92.6±0.7   | 83.2±0.6  | 30.1±1.5  |
> | relocate-cloned-v1  | 38.5±0.3  | 25.6±0.6  | 6.2±0.7  |
> | antmaze-medium-play-v2  | 99.7±0.6   | 84.1±0.3  | 83.3±1.2|
> | antmaze-medium-diverse-v2  | 99.1±0.6   | 87.1±0.3| 72.4±0.9 |
> | antmaze-large-play-v2  | 92.5±1.3| 81.0±1.6  | 67.6±0.6 |
> | antmaze-large-diverse-v2  | 99.1±1.6   | 72.7±1.2  | 70.2±1.5 |
>
> >Lack of description about how to build and train w in the implementation. Moreover, computational cost analysis for the InPG module are missing.
>
> Thank you for pointing that out. The w network is initialized as a value function and trained using the loss defined in Equation (7). The learning rate and network architecture are provided in the appendix. For clarity, we include a code snippet below demonstrating the training of the w network, which only requires **less than 20** lines beyond the original IVR implementation.
>
> ```
> # grad of q to pi
> pi, _ = self.actor.sample(observations, deterministic=False)
> q_pi = self.q(observations, pi)
> q_grad_pi = torch.autograd.grad(
>     q_pi.mean(),
>     self.actor.parameters(),
>     retain_graph=True
> )
> q_grad_pi_flat = torch.cat([grad.view(-1) for grad in q_grad_pi], dim=0)
> w = self.w(observations, actions)
> bc_losses = self.actor.evaluate(observations, actions)
> policy_loss = torch.mean(w * bc_losses)
> # grad of wbc to pi
> wbc_grad_pi = torch.autograd.grad(
>     policy_loss,
>     self.actor.parameters(),
>     retain_graph=True,
>     create_graph=True,
> )
> wbc_grad_pi_flat = torch.cat([grad.view(-1) for grad in wbc_grad_pi], dim=0)
> # w loss
> w_loss = F.mse_loss(q_grad_pi_flat, wbc_grad_pi_flat)
>
> ```
>
> The add of the InPG module doesn't add too much computational cost, as shown below.
>
> |  Offline Setting      | IVR | Uni-IVR
> | --------------- | ----| ----|
> |Run Time (training+evaluation) | ~5h| ~6h|
>
>
> >The paper proposes two alternative objectives that enable the use of Flow-Matching and Diffusion models instead of traditional Gaussian policies. However, if I understand correctly, these objectives treat all actions from the dataset uniformly and do not incorporate information from the Q and V functions.
>
> There seems to be a misunderstanding here. Note that our proposed two alternative objectives are intended to replace the log-likelihood maximization objective used for training Gaussian policies, as mentioned in lines 240–241. In other words, we are replacing the standard objective $\max E_D(\log \pi(a|s))$ with our alternative formulations. However, the learning of the w function remains essential for performing weighted behavior cloning. We will clarify this more explicitly in the revised version of the paper.
>
>
> >in Figure 2, the authors provide Q-curves to illustrate higher performance; however, Q-learning methods (IQ-learn [1] for example) can produce high Q-values without corresponding performance gains. There are no explanation for these Q value results.
>
> Note that we provide the corresponding evaluation scores in the first row of Figure 2, and the Q-value is included only as an additional metric for reference. As shown in Figure 2, the actual performance aligns well with the Q-values.
> In fact, Q-value overestimation is a much more serious issue in the offline setting. In the online setting, the Q-value can still serve as a useful performance indicator—though it may exhibit slight overestimation due to TD learning, it is not nearly as severe as in the offline case.
>
> >In line 298-299, author stated that “DICE-based methods only work for the offline setting”. Can you justify the reason why they can not be applied by the reference policy?
>
> Thanks for pointing out that, the reason is that if we apply DICE-based methods to the online setting, sampling from the stationary distribution $d^{\mu}(s, a)$ of the reference policy is needed.
>
> More specifically, this is the dice learning objective:
> $$
> \pi^{*} = \arg \max_{\pi} E_{(s, a) \sim d^{\pi}}[r(s, a)] - \alpha D_f(d^{\pi}||d^{\mu}).
> $$
> Solve by duality we get
> $$
> \min_{V}  (1 - \gamma)E_{s \sim d_{0}}\left[ V(s) \right] + \alpha E_{(s, a) \sim d^{\mu}} \big[ f_{dual}\big(\left[T_r V(s, a) - V(s)\right] / \alpha\big) \big],
> $$
> Samples from $d^{\mu}(s, a)$ are intractable in the online setting where $\mu$ is the previous policy.
>
> >This raises questions about the sensitivity of the method to this hyperparameter. Could you provide an ablation study that shows how performance varies with λ in each setting?
>
> Due to space limit, please see the response to Reviewer kppM for ablation study of different $\lambda$.
>
>
> >To support the claim of scalability, could you please add the results for a Flow-matching policy to Table 1, as was done for the Diffusion policy?
>
> |  Offline setting      | Uni-IVR (diffusion) | IVR (flow-matching) |
> | --------------- | ---- | --- |
> | halfcheetah-m  | 58.0 ±0.6  | 56.5 ±1.4  |
> | hopper-m  | 102.5 ±0.2 | 100.6 ±2.2  |
> | walker2d-m  | 92.3 ±0.1 | 91.8 ±1.4  |
> | halfcheetah-m-r  | 48.4 ±0.9 | 49.7 ±0.4  |
> | hopper-m-r  | 101.3 ±2.1 | 101.8 ±1.1  |
> | walker2d-m-r  | 90.8±1.6 | 92.8±1.3 |
> | halfcheetah-m-e  |  97.3 ±0.6 | 97.4 ±0.6 |
> | hopper-m-e  | 111.2 ±0.3 | 112.6 ±0.4  |
> | walker2d-m-e  | 114.1±0.5 | 113.7±0.4  |
> | antmaze-u  | 98.1 ±1.8 | 97.3 ±1.0  |
> | antmaze-u-d  | 82.0 ±1.4 | 83.5 ±1.8  |
> | antmaze-m-p  | 88.5 ±3.1 | 89.6 ±4.5  |
> | antmaze-m-d  | 89.7 ±2.8 | 86.7 ±1.1  |
> | antmaze-l-p  | 68.6 ±3.6 | 72.4 ±2.2  |
> | antmaze-l-d  | 69.0 ±4.5 | 74.2 ±3.8  |
>
>
> We believe our clarifications to your responses naturally answered your questions listed. Please let us know if any further questions remain. We hope the reviewer can reassess our work with these clarifications.

---

> > ### Comment · Reviewer_HJMu · 2025-08-05
> >
> > I would like to thank the authors for their detailed rebuttal. While I still have some concerns regarding the implementation and reproducibility, I acknowledge that this work makes a valuable contribution to the RL community. Therefore, I will raise my score.

---

### Official Review · Reviewer_kppM · 2025-07-02

**Clarity:** 3
**Significance:** 3
**Originality:** 4
**Rating:** 5
**Confidence:** 4

**Summary:**

This paper proposes Uni-IVR, a unified model-free Reinforcement Learning (RL) framework. Uni-IVR  builds upon the Implicit Value Regularization (IVR) framework, and is designed to address online, offline, and offline-to-online settings. It resolves the issue of over-conservatism when directly applying IVR in online settings, and provides an implicit trust-region-style update via the value function. Furthermore, Uni-IVR introduces a unified policy extraction objective that theoretically guarantees lower value estimation error and greater performance improvement relative to the reference policy. Evaluations on 6 RL benchmarks and 23 environments demonstrate that Uni-IVR achieves competitive or superior performance compared to state-of-the-art baselines across all three settings.

**Questions:**

-

**Ethical Concerns:**

["NO or VERY MINOR ethics concerns only"]

**Final Justification:**

The paper makes a contribution to the field of reinforcement learning, with good writing, methodology, and experiments. The authors have also provided timely and thorough responses to the reviewers’ comments. I recommend accepting this paper.

**Limitations:**

1. The paper does not provide detailed information on the computational resources used for the experiments.
2. Although the paper considers various learning scenarios such as online, offline, and offline-to-online, it is limited to model-free RL. Verification of the proposed method in other settings, such as model-based RL, could further demonstrate its generality.

**Quality:**

4

**Strengths And Weaknesses:**

## Strengths
1. Uni-IVR provides a unified model-free RL framework that integrates online, offline, and offline-to-online learning settings, demonstrating strong versatility.
2. The paper offers theoretical guarantees, including lower value estimation error and greater performance improvement for its unified policy extraction objective relative to the reference policy. It also provides theoretical insights, indicating that InPG (In-sample Policy Gradient) optimizes the constrained learning objective, can better utilize the Q-function, and can achieve a greater performance improvement than $\pi_{IVR}^*$.
3. Uni-IVR achieves higher sample efficiency in online RL compared to both off-policy methods without trust-region updates and on-policy methods with trust-region updates.

## Weaknesses
1. The algorithms compared in the online RL experiments are outdated (the latest is from 2018); it should consider comparing with more advanced algorithms.
2. The "iteratively refined behavior policy" used to address the over-conservatism issue in online IVR is a soft update (Equation 5). While more lightweight than dataset filtering, the choice of hyperparameter λ might affect performance.

---

> ### Author Rebuttal · Authors · 2025-07-31
>
> We thank the reviewer for the effort engaged in the review phase and the constructive comments. Regarding the concerns, we provide the detailed responses separately as follows.
>
> >The algorithms compared in the online RL experiments are outdated (the latest is from 2018); it should consider comparing with more advanced algorithms.
>
> Thank you for your suggestions. The main contribution of our paper is the proposal of a unified framework for off-policy RL. To highlight Uni-IVR’s scalability across different settings, we select representative baselines from online, offline, and offline-to-online RL. To further demonstrate Uni-IVR’s competitiveness in the online setting, we compare it with BAC [1], which, to the best of our knowledge, is the strongest model-free, off-policy online RL algorithm in recent years. Using the official implementation of BAC, our results show that Uni-IVR consistently outperforms it across all tasks.
>
>
> |  Algorithm      | Uni-IVR | BAC |
> | --------------- | ---- | --- |
> | Hopper-v3  | **3683**±15  | 3450±10    |
> | HalfCheetah-v3  | **14305**±25 | 13110±22  |
> | Humanoid-v3  | **5820**±27 | 5040±10    |
> | Walker2d-v3  | **5940**±16 | 5540±31    |
> | finger-turn_hard  | **625**±80 | 327±49    |
> | quadruped-walk  | **924**±14 | 457±22    |
>
>
>
>
>
> >The "iteratively refined behavior policy" used to address the over-conservatism issue in online IVR is a soft update (Equation 5). While more lightweight than dataset filtering, the choice of hyperparameter λ might affect performance.
>
> Thank you for raising this point. We provide full ablation results for different values of $\lambda$ across all environments (with * marking the $\lambda$ values we selected and their corresponding results reported in the paper). The optimal range of $\lambda$ **varies across settings**, as the desired contribution of the previous policy differs. In the online setting, a smaller $\lambda$ is preferred because earlier policies are generally suboptimal. In contrast, in the offline-to-online setting, a larger $\lambda$ is beneficial due to the availability of a strong initial policy.
>
> Nonetheless, within the same setting, the performance is **relatively robust** to the choice of $\lambda$, as shown in the table below. We note that we did not explore a broader range of values, which might yield even better results.
>
>
> |   Online     | $\lambda=0.01$ | $\lambda=0.05$ | $\lambda=0.1$ | $\lambda=0.2$ |
> | --------------- | ---- | --- | --- |--- |
> | Hopper-v3  | 2674±24  | 3034±18  | 3628±15  | 3683±15 (\*) |
> | HalfCheetah-v3  | 11785±19 | 13943±17  | 14331±23  | 14305±25 (\*) |
> | Humanoid-v3  | 4124±27 | 5423±19  | 5689±22  | 5820±27 (\*) |
> | Walker2d-v3  | 5332±23 | 5734±15  | 5966±20  | 5940±16 (\*) |
> | finger-turn_hard  | 336±56 | 595±64  | 625±80 (\*) | 612±54 |
> | quadruped-walk  | 746±16 | 902±12  | 924±14 (\*) | 954±17 |
> | AntBulletEnv-v0  | 2393±13 | 3683±23  | 2892±12 (\*) | 2692±19 |
> | BipedalWalker-v3  | 167±16 | 271±11  | 301±16 (\*) | 236±18 |
>
> |   Offline-to-Online     | $\lambda=0.01$ | $\lambda=0.05$ | $\lambda=0.1$ | $\lambda=0.2$ |
> | --------------- | ---- | --- | --- |--- |
> | pen-cloned-v1  | 88.5±0.4  (\*)  |86.6±0.5  | 83.5±0.6  | 82.7±1.6 |
> | door-cloned-v1  | 93.3±1.9  (\*) | 90.3±1.2  | 91.6±1.4  | 87.9±0.6 |
> | hammer-cloned-v1  | 92.6±0.7  (\*) | 93.2±0.6  | 90.1±1.5  | 88.1±0.4|
> | relocate-cloned-v1  | 38.5±0.3  (\*) | 35.6±0.6  | 36.2±0.7  | 32.5±0.2|
> | antmaze-medium-play-v2  | 99.7±0.6  (\*) | 97.1±0.3  | 93.3±1.2| 90.7±1.4 |
> | antmaze-medium-diverse-v2  | 99.1±0.6  (\*) | 97.1±0.3| 92.4±0.9  |92.3±1.0|
> | antmaze-large-play-v2  | 92.5±1.3 (\*) | 91.0±1.6  | 87.6±0.6 | 85.1±0.9 |
> | antmaze-large-diverse-v2  | 99.1±1.6  (\*) | 92.7±1.2  | 90.2±1.5 | 89.7±0.7 |
>
>
> >The paper does not provide detailed information on the computational resources used for the experiments.
>
> Thanks for pointing out that, we use NVIDIA A40 for all experiments.
>
>
> >Although the paper considers various learning scenarios such as online, offline, and offline-to-online, it is limited to model-free RL. Verification of the proposed method in other settings, such as model-based RL, could further demonstrate its generality.
>
> Thanks for pointing out that, the value and policy learning in Uni-IVR can be naturally extended to dyna-style model-based RL algorithms like MBPO [2] where the standard Q-value update rule is replaced by ours. We leave it as future work.
>
> [1] [Seizing Serendipity: Exploiting the Value of Past Success in Off-Policy  Actor-Critic](https://arxiv.org/pdf/2306.02865), Ji et al, ICML 2024.
>
> [2] [When to Trust Your Model: Model-Based Policy Optimization](https://arxiv.org/pdf/1906.08253), Janner et al, NeurIPS 2019.

---

> > ### Comment · Reviewer_kppM · 2025-08-06
> >
> > Thank you for addressing my comments thoroughly, and I appreciate the authors’ efforts. Hopefully my comments can enrich your work.

---

### Public Comment · ~Zhihan_Liu1 · 2025-12-05
**Missing Discussions with Highly Relevant Previous Work**

Dear Authors,
We came across your poster and found your work very interesting and impressive. However, we noticed that the paper does not discuss highly relevant prior NeurIPS works ([1][2]), which also propose value-regularized methods to achieve optimism in the online setting ([1]) and pessimism in the offline setting ([2]) for provable sample efficiency and empirical performance gain. We believe that adding a discussion of [1][2] and clarifying the relationship between your approach and theirs in the camera-ready version would further strengthen the paper.

[1] Liu, Zhihan, et al. "Maximize to explore: One objective function fusing estimation, planning, and exploration." Advances in Neural Information Processing Systems 36 (2023): 22151-22165.
[2] Liu, Zhihan, et al. "Provably mitigating overoptimization in rlhf: Your sft loss is implicitly an adversarial regularizer." Advances in Neural Information Processing Systems 37 (2024): 138663-138697.

---

### Decision · Program_Chairs · 2025-09-17

**Decision:**

Accept (poster)

**Comment:**

This paper introduces Uni-IVR, a framework that aims to use a single formulation for various reinforcement learning setups, such as online RL, offline RL, and offline-to-online RL. Reviewers highlighted multiple strengths of the paper: the paper tackles an important problem based on a well-presented theoretical analysis and provides strong empirical results. I agree with these points. Overall, this paper is technically solid and achieved the ambitious goal of being competitive on various setups with a unified setup.

After carefully reading the paper, reviews, and discussion, I recommend accepting the paper. Initially there were some concerns raised by reviewers such as the lack of comparison to recent baselines or hyperparameters, but the authors’ rebuttal successfully addressed these concerns by providing additional experimental results. While only one reviewer ended up in a non-borderline acceptance recommendation, all the reviewers have no remaining concerns as the result of active discussion with authors. It is encouraged that the authors release open-source implementation by following the suggestion of Reviewer HJMu, incorporate all the points raised by reviewers, and provide confidence intervals as pointed out by Reviewer wWsA.